**Data Availability Statement:** The raw RNAseq reads produced in this study have been deposited on the NCBI sequence read archive database with the Bioproject accession PRJNA627587 under this

# Transcriptome and proteome of the corm, leaf and flower of *Hypoxis hemerocallidea* (African potato)

Mihai-Silviu Tomescu[1], Selisha Ann Sooklal[2], Thuto Ntsowe[3], Previn Naicker[4], Barbara Darnhofer[5,6,7], Robert Archer[8], Stoyan Stoychev[4], Dirk Swanevelder[3], Ruth Birner-Grünberger[5,6,7¤], Karl Rumbold🖂[1] *

**1** School of Molecular and Cell Biology, University of the Witwatersrand, Johannesburg, South Africa, **2** Department of Life and Consumer Sciences, College of Agriculture and Environmental Sciences, UNISA, Johannesburg, South Africa, **3** Biotechnology Platform, Agricultural Research Council, Onderstepoort, South Africa, **4** Council for Scientific and Industrial Research, Pretoria, South Africa, **5** ACIB GmbH, Graz, Austria, **6** Institute for Pathology, Medical University of Graz, Graz, Austria, **7** Omics Center Graz, BioTechMed, Graz, Austria, **8** National Herbarium, South African National Biodiversity Institute, Pretoria, South Africa

¤ Current address: Institute of Chemical Technologies and Analytics, TU Wien, Vienna, Austria
* karl.rumbold@wits.ac.za

## Abstract

The corm of *Hypoxis hemerocallidea*, commonly known as the African potato, is used in traditional medicine to treat several medical conditions such as urinary infections, benign prostate hyperplasia, inflammatory conditions and testicular tumours. The metabolites contributing to the medicinal properties of *H. hemerocallidea* have been identified in several studies and, more recently, the active terpenoids of the plant were profiled. However, the biosynthetic pathways and the enzymes involved in the production of the terpene metabolites in *H. hemerocallidea* have not been characterised at a transcriptomic or proteomic level. In this study, total RNA extracted from the corm, leaf and flower tissues of *H. hemerocallidea* was sequenced on the Illumina HiSeq 2500 platform. A total of 143,549 transcripts were assembled *de novo* using Trinity and 107,131 transcripts were functionally annotated using the nr, GO, COG, KEGG and SWISS-PROT databases. Additionally, the proteome of the three tissues were sequenced using LC-MS/MS, revealing aspects of secondary metabolism and serving as data validation for the transcriptome. Functional annotation led to the identification of numerous terpene synthases such as nerolidol synthase, germacrene D synthase, and cycloartenol synthase amongst others. Annotations also revealed a transcript encoding the terpene synthase phytoalexin momilactone A synthase. Differential expression analysis using edgeR identified 946 transcripts differentially expressed between the three tissues and revealed that the leaf upregulates linalool synthase compared to the corm and the flower tissues. The transcriptome as well as the proteome of *Hypoxis hemerocallidea* presented here provide a foundation for future research.

link: https://www.ncbi.nlm.nih.gov/bioproject/PRJNA627587. Data are also available in PRIDE under the following details: Hypoxis hemerocallidea (Afrtican potato) corm, leaf and flower transcriptomics and proteomics ProteomeXchange accession: PXD019784 http://www.ebi.ac.uk/pride/archive/projects/PXD019784.

**Funding:** Our gratitude goes to the Department of Science and Technology, South Africa, for the Biocatalysis Initiative. MST received a PhD bursary from the National Research Foundation, South Africa, grant number 102123. The funders (Omics Center Graz, BioTechMed and ACIB GmbH) provided support in the form of salaries for authors [B.D and R.B.G], but did not have any additional role in the study design, data collection and analysis, decision to publish, or preparation of the manuscript. The specific roles of these authors are articulated in the 'author contributions' section.

**Competing interests:** The funders (Omics Center Graz, BioTechMed and ACIB GmbH) provided support in the form of salaries for authors [B.D and R.B.G]. This does not alter our adherence to PLOS ONE policies on sharing data and materials.

## Introduction

The term *African potato* refers to the tuberous rhizome of the herbaceous plant species belonging to the Hypoxidaceae family. There are about 90 species belonging to the *Hypoxis* genus. One of the more popular species is *Hypoxis hemerocallidea*. Noteworthy, *H. obconica*, *H. patula* and *H. rooperi* have been identified as synonymous species [1] and will here forth be commonly referred to as *H. hemerocallidea*.

The rhizomes of *Hypoxis spp.* are used in traditional medicine to treat several illnesses such as urinary infections, inflammatory conditions, hypertension, testicular tumours, some cancers, and HIV-AIDS [2]. Having attracted sufficient attention for its medicinal properties, in 1969 a hydroalcoholic extract from *H. hemerocallidea* was patented for anti-inflammatory, antibiotic, anti-arthritic, anti-atherosclerotic and diuretic properties, as well as a stimulant of muscular and hormonal activities [3]. Later research has corroborated some of the medicinal properties. Aqueous and methanolic extracts showed anti-inflammatory effects in rats with induced edema in the paw by subplantar injections with fresh egg albumin [4]. Aqueous extracts were found to have antinociceptive properties, antidiabetic properties [5, 6] and delayed the onset of seizures induced with pentylenetetrazole (PTZ).

Some of the metabolites found within *Hypoxis spp.* have shown chemical and biological activity. The most abundant of which, hypoxoside (hydrolysed to form rooperol), was shown to be a powerful anti-oxidant [7]. In addition, rooperol has shown anti-BL6 melanoma activity in rats [8]. Over the counter preparations containing *Hypoxis* phytosterols and β-sitosterols have been distributed for the treatment of benign prostate hyperplasia and as immune-system boosters [9].

The chemical synthesis of hypoxoside, rooperol and rooperol-derivatives have been patented [10] alongside their use in the treatment of inflammation [11, 12] and viral infections [13]. However, the synthetic production of hypoxoside has been documented to be difficult [14]. Moreover, the cultivation of *H. hemerocallidea* is known to be problematic due to lengthy seed dormancy [15], and tissue culture of *H. hemerocallidea* produces low yields of hypoxoside rendering these methods impractical for the extraction of certain active metabolites [14]. Research into the medicinal application of the African potato has also indicated that beneficial effects of the plant are largely dependent on the harvesting season. For example, African potato harvested in autumn and winter displays improved antimicrobial properties against *Bacillus subtilis*, *Escherichia coli*, *Klebsiella pneumonia*, *Staphylococcus aureus* [16]. This leaves an alternative, important and yet unexplored gap in the biocatalytic production of hypoxoside, rooperol and other important metabolites which can circumvent the need for cultivation or tissue culturing of *Hypoxis hemerocallidea*. Another important medicinally active compound found within the African potato is stigmasterol. However, this compound can be produced on an industrial scale from *Chlorophytum borivilianum* [17]. Nevertheless, it is unlikely that the medicinal benefits provided by the African potato are solely due to rooperol and stigmasterol [18]. There are numerous other secondary plant metabolites that could contribute to the medicinal properties of the African potato. Terpenoids, saponins, cardiac glycosides, tannins and reducing sugars have all been detected in the African potato [19, 20]. Not all secondary metabolites have been characterised though, and even less so are the enzymes participating in the biosynthetic pathways to produce the secondary metabolites within the plant. The African potato is practically undocumented at genomic, transcriptomic and proteomic levels, having only 11 nucleotide sequences on the NCBI database (https://www.ncbi.nlm.nih.gov/search/?term=hypoxis+hemerocallidea accessed on: 15th April 2020). There is a notable gap in the molecular information available for the African potato regarding the -omics as well as the characterisation of the enzymes and pathways. In this study, the first comprehensive

transcriptome of *H. hemerocallidea* was assembled *de novo* and functionally annotated providing a useful resource of genetic information for downstream research. Further, differential expression analysis between the corm, leaf and flower provides a reduced set of uncharacterised genes that narrows down the list of genes possibly imparting medicinal properties to the corm. Cross-tissue transcriptomic analyses have been performed in the past to identify candidate genes involved in the biosynthesis of secondary metabolites in medicinal plants like *Ferula asafoetida*, *Dysphania schraderiana* and *Salvia miltiorrhiza* [21–23]. In this study, aside from transcriptomic analyses, proteomic profiling was performed on the flower, leaf and corm tissues from *Hypoxis hemerocallidea*.

# Methods and materials

## Plant material collection, storage and preparation

An approximately two-year-old *Hypoxis hemerocallidea* (African potato) plant, grown under natural conditions, was identified and collected at the Pretoria National Botanical Gardens (South Africa) under the expertise of Dr. Robert Archer. Biological replicates of the flower and leaf material were cut and stored in 50 ml centrifuge tubes. The flower material included the sepal and receptacle but not the stamen. The samples were immediately frozen in liquid nitrogen and stored at -80˚C until use. The corm was washed with distilled water, cut into cubes of approximately 2 cm$^3$, frozen in liquid nitrogen and stored at -80˚C until use. The lack of *H. hemerocallidea* specimens, restricted the experimental setup to technical replicates for the corm tissue. Unless otherwise stated, plant material was routinely crushed into a fine powder in liquid nitrogen using a sterile mortar and pestle.

## Extraction of total RNA and sequencing using the Illumina Hi-Seq 2500 platform

Total RNA from the corm, leaf and flower of *H. hemerocallidea* was extracted in duplicate using the Trizol® reagent from Sigma-Aldrich (Massachusetts, USA) according to the manufacturer's instructions. Total RNA was quantified using the Qubit Fluorometer 2.0 from Life Technologies (California, USA). Transcriptome (cDNA) library preparation and sequencing were conducted at the Agricultural Research Council Biotechnology Platform (Pretoria, South Africa). Samples were depleted of ribosomal RNA using the Ribo-Zero Plant rRNA Removal Kit from Illumina (California, USA) according to the manufacturer's instructions. Libraries for the corm (technical duplicates), leaf and flower (biological duplicates) were created and tagged with index adaptors for multiplex sequencing using the TruSeq Stranded mRNA Library Preparation Kit (Illumina, California, USA). Samples, multiplexed with transcriptome samples from *Helianthus annuus* (Sunflower), were subjected to paired-end sequencing on the Illumina Hi-Seq 2500 platform using the Illumina Hi-Seq Reagent Kit v4 from Illumina (California, USA).

## Quality control and trimming of low-quality reads

The quality of the reads before and after trimming was assessed with FastQC version 0.11.5 [24]. Trimming was performed with Trimmomatic version 0.36 [25]. In brief, Illumina adapters (TruSeq3-PE-2.fa:2:30:10) were removed along with leading bases with a quality below 7 and trailing bases with a quality below 10. Trimming was performed on a sliding window of 4 bases, and were trimmed if any of the 4 bases had a quality below a Phred score of 15. Moreover, sequence reads with a length below 36 were discarded. Satisfactory trimming of the reads was identified by a resultant average Phred score above 28.

## De novo assembly of the *Hypoxis hemerocallidea* transcriptome

The high-quality paired-end reads of the flower, leaf and corm tissue of *H. hemerocallidea* were concatenated and assembled *de novo* (in the absence of a reference genome) into a single RNA-seq dataset using Trinity version 2.6.6 under default settings [26, 27]. The assembly serves as a reference transcriptome for downstream analyses.

## Identification and removal of contaminant isoforms

The raw reads generated in this study for *H. hemerocallidea* were obtained following multiplex sequencing with *Helianthus annuus* (Sunflower). As such, the *H. hemerocallidea* transcriptome was analysed by blastn search for cross contamination against the reference genome of *H. annuus* (accession number: han_ref_HanXRQr1_0) obtained from RefSeq using BLAST + 2.7.0 with an e-value of 1e-20 [28]. The number of hits were plotted against percentage identity obtained from the blastn outfmt6 results at one percent intervals. Likewise, the decontaminated transcriptome was also assessed for the presence of contaminants.

Decontamination was performed using DeconSeq [29]. The genome of *Helianthus annuus* was used to identify contaminants. The genomes of *Elaeis guineensis*, *Musa acuminate*, *Prunus persica* and *Vitis vinifera* were used as databases for retaining non-contaminating sequences as those species were found to be the most similar species annotated on the NCBI nr database [30]. S1 Table provides the RefSeq accession numbers of all the genomes used in this study. Only clean isoforms were retained. The clean isoforms were searched against the top 9 similar species and *Helianthuus anuus* to verify efficiency of the decontamination procedure (S2 Table).

## Functional annotation of assembled transcript isoforms

Assembled transcripts were searched against the NCBI non-redundant (nr) [30], Gene Ontology (GO) [31], Protein Families (Pfam) [32] databases and enzyme commission (EC) numbers were assigned using FunctionAnnotator [33] in order to assign putative function and assess taxonomic distribution. Noteworthy, FunctionAnnotator makes use of Blast2GO [34, 35] to annotate sequences on the GO database. Annotation of transcripts was also performed on the Swiss-Prot database [36] using Trinotate version 3.2.0 [37]. Transcripts were additionally searched against the evolutionary genealogy of genes: Non-supervised Orthologous Groups (eggNOG) database using eggNOG mapper version 5.0 [38]. In this instance, annotations were assigned single-letter codes to classify descriptions into 25 broad functional groups based on Clusters of Orthologous Groups (COG). Pathway identification was performed by searching transcripts against the Kyoto Encyclopaedia of Genes and Genomes (KEGG) database using KOBAS 3.0 [39, 40]. Annotation of transcription factors was performed on the Plant Transcription Factor Database 2.0 (PlantTFDB) [41]. Open reading frames (ORFs) were predicted using TransDecoder version 5.2.0 [27]. *Hypoxis hemerocallidea* is classified under the Asparagales order. However, there were no species from the Asparagales order present within the top 10 similar species. To get an insight into the evolutionary relatedness of *H. hemerocallidea* to species from the Asparagales order, a blastn search was performed against the transcriptomes of *Asparagus officinalis* (garden asparagus) (RefSeq: GCF_001876935.1), *Dendrobium catenatum* (RefSeq: GCA_001605985.2 and *Phalaenopsis equestris* (RefSeq: GCF_001263595.1) (from the Asparagales order). The top 6 similar species were included in the analyses as well.

## Differential expression analysis

Alignment-based isoform abundance estimation was performed using RNA-Seq by Expectation Maximization (RSEM) version 1.3.1 [42]. In brief, the raw reads of each tissue were

aligned with Bowtie to the assembled transcriptome and the relative abundance of transcripts in each tissue was estimated with RSEM. Differential transcript expression analysis between the corm, leaf and flower tissues of *H. hemerocallidea* was performed using edgeR [43] with $\log_2(\text{fpkm}+1)$ normalisation. Differential expression analysis was performed to identify transcripts expressed in significantly elevated levels which could possibly confer some of the phytomedicinal properties associated with the corm and leaf tissues. The workflow was implemented using the Trinity pipeline [27] with a cut-off p-value set to 0.05. Clustering of expressed transcripts between the sample replicates from the leaf and flower, and technical replicates from the corm tissue was performed hierarchically using normalised transcript expression with edgeR. Transcripts are clustered together by similar expression levels.

## Proteomic characterisation

Various extraction and library preparation approaches were used to maximise the detection of proteins from *H. hemerocallidea*. The first approach involved the extraction of total proteins (in triplicate) under denaturing and reducing conditions from the corm, leaf and flower tissues. The proteins were subjected to on-particle digestion with trypsin and, thereafter, LC-MS/MS proteomic analysis. The second approach set out to identify proteins that could be extracted in a soluble state from the corm, leaf and flower tissues using a commercial plant protein extraction kit. The third approach, applied exclusively to the corm tissue, was used to enrich low abundant soluble proteins through ion-exchange chromatography. Proteins obtained from the second and third methods were subjected to in-gel digestion with trypsin and, thereafter, LC-MS/MS proteomic analysis.

**Protein extraction under denaturing and reducing conditions and on-particle digestion with trypsin.** Crushed corm, leaf and flower tissues were solubilised in 50 mM sodium borate buffer pH 8.5 containing 4% SDS and 100 mM DTT. Samples were heated at 95°C for 5 minutes, thereafter, samples were sonicated at 50% amplitude using a QSonica Q125 sonicator (10 s bursts, 30 s rest for 5 cycles on ice) and centrifuged at 14,000 x g for 30 minutes at room temperature. Proteins were then precipitated with 4 volumes of 12.5% TCA/acetone at -20°C. Precipitated protein was pelleted by centrifugation at 14,000 x g at 4°C, washed by resuspension in ice-cold 80% acetone and subsequently centrifuged again. Washed pellets were resuspended in 20 mM Tris-HCl buffer at pH 8.0 containing 4% SDS. Cysteine residues were then reduced with 10 mM DTT at 37°C for 30 minutes. Subsequently, cysteine residues were alkylated by incubating with 40 mM iodoacetamide (IAA) at 37°C in the dark. IAA was quenched with 20 mM DTT. Sample clean-up was carried in LoBind tubes from Eppendorf (Hamburg, Germany) by adding 20 mg/ml MagReSyn® HILIC beads solution in a 1:10 protein to beads solution ratio. Prior to the addition of protein sample, the beads were equilibrated with 200 μl of 100 mM ammonium acetate equilibration buffer at pH 4.5 containing 15% (v/v) acetonitrile. The equilibration step was repeated 3 times, while the removal of the supernatant was performed with the aid of a magnetic stand. After mixing protein and beads, binding buffer (200 mM ammonium acetate, 30% acetonitrile, pH 4.5) was added to a final concentration of 15% acetonitrile and 100 mM ammonium acetate. The protein-beads mixture was then mixed at room temperature for 30 minutes using an Intelli Mixer from ELMI (Riga, Latvia) set on the UU mode at 30 RPM. Thereafter, the supernatant was removed, and beads were washed twice for 1 minute in 200 μl 95% acetonitrile. On-particle digestion was performed for 4 hours at 37°C in 50 mM ammonium bicarbonate pH 8.0 with a 1:10 ratio of trypsin from Sigma-Aldrich (Missouri, USA) to protein.

**LC-MS/MS analysis of on-particle digested proteins.** Approximately 1 μg of peptides generated by digestion with trypsin were de-salted inline using an Acclaim PepMap trap

column (C18, 3μm, 20m × 0.075 mm) for 2 minutes at 5μl/min in 2% acetonitrile and 0.2% formic acid. Trapped peptides were then separated using an Acclaim PepMap RSLC column (C18, 2 μm, 150 × 0.075 mm) through the Dionex Ultimate 3000 RSLC system at a flow rate of 0.5μl/min. For separation, peptides were eluted with a gradient of 4–40% B over 60 minutes where A is 0.1% formic acid and B is 80% acetonitrile with 0.1% formic acid. Mass spectrometry analysis was performed using an AB Sciex 6600 TripleTOF mass spectrometer operated in positive ion mode. Data-dependent acquisition (DDA) was employed for MS data. Precursor MS scans were acquired from $m/z$ 400–1500 ($2^+$– $5^+$ charge states) using an accumulation time of 250 ms flowed by 80 fragment ion (MS/MS) scans, acquired from $m/z$ 100–1800 with 25 ms accumulation time each. Raw data files were searched with the Protein Pilot software (SCIEX), using a database containing 6-frame translated sequences from the assembled *H. hemerocallidea* transcriptome as well as common contaminants. Trypsin was set as the digestion enzyme, cysteine alkylation (iodoacetamide) was allowed as a fixed modification and biological modifications allowed in the search parameters. Protein identification was restricted to proteins with 2 unique peptides or more.

**Protein extraction using P-PER.** The working solution (WS) from the P-PER plant protein extraction kit (Thermo Scientific, Massachusetts, USA) was prepared according to the manufacturer's instructions with the addition of 10 mm DTT and cOmplete ULTRA protease inhibitor cocktail tablets from Roche (Basel, Switzerland). *H. hemerocallidea* plant material (corm, leaf and flower) was individually crushed in liquid nitrogen using a sterile mortar and pestle. The WS was then mixed with 80 mg of crushed plant tissue in the provided polypropylene mesh bags, in which the mixture was further homogenised mechanically. The homogenous mixture was centrifuged for 5 minutes at 5,000 x *g* at room temperature. The lower aqueous layer, containing the extracted soluble proteins, was transferred to a clean Eppendorf tube. The protein was quantified using the Qubit Fluorometer 2.0 from Life Technologies (California, USA) and then subjected to electrophoresis as described in the "Sodium-dodecyl sulphate polyacrylamide gel electrophoresis" section.

**Corm protein extraction and fractionation.** The corm from *H. hemerocallidea* is a tuberous tissue with a high starch content. This inherently results in low protein yields following extraction. In order to enrich proteins from the corm tissue, fractionation was employed. Approximately 5 g of *H. hemerocallidea* corm tissue was crushed in liquid nitrogen using a sterile mortar and pestle. The ground tissue was resuspended in 5 volumes of 50 mM sodium borate buffer pH 9.0 containing 5 mM DTT, 5% PVPP and a cOmplete ULTRA protease inhibitor cocktail tablet from Roche (Basel, Switzerland). Protein was extracted in the aforementioned buffer with gentle stirring at 4°C for 1 hour. The slurry was then centrifuged at 20,000 x *g* for 30 minutes at 4°C. The supernatant was loaded onto a 5 ml HiTrap DEAE FF column from GE Healthcare (Illinois, USA), pre-equilibrated with 50 mM sodium borate buffer pH 9.0. Proteins were eluted over 50 ml using a linear gradient from 0 M to 1 M NaCl in 50 mM sodium borate buffer pH 9.0. Fractions of 5 ml each were collected. Chromatography was carried out on the ÄKTA Prime Plus from GE Healthcare (Illinois, USA) with the flow rate maintained at 5 ml/min. Fractions containing protein were then extracted in tris-buffered phenol and precipitated with ice-cold 0.1 M ammonium acetate in methanol. Samples were washed with a ice-cold solution of 10 mM DTT in acetone and air-dried. The protein pellets were resuspended in reducing sample buffer and subjected to electrophoresis as described in the "Sodium-dodecyl sulphate polyacrylamide gel electrophoresis" section. Protein concentrations were routinely determined using the Qubit Fluorometer 2.0 from Life Technologies (California, USA).

**Sodium-dodecyl sulphate polyacrylamide gel electrophoresis.** Precipitated protein samples were re-suspended in a 3:1 ratio of protein to reducing sample buffer (150m mM Tris-

HCl at pH 7.0, containing 12% SDS (w/v), 6% β-mercaptoethanol (v/v), 0.05% Coomassie blue G-250 and 30% (w/v) glycerol) [44]. Solubilised samples were electrophoresed for 0.4 mm distance through 8% acrylamide gels to remove salts and small metabolites. Thereafter, 0.3 mm gel pieces (stained with Coomassie G250) were excised and further prepared for LC-MS/MS analysis. Gels were fixed and stained in staining solution (0.025% Coomassie G-250, 40% methanol, 10% acetic acid prepared with milliQ). Thereafter de-stained once in de-stain solution 1 (40% methanol and 10% acetic acid prepared with milliQ) and twice in 10% acetic acid. Excised gel pieces were stored in 10% ethanol until in-gel extraction. To prevent un-polymerised acrylamide from forming adducts with electrophoresed proteins, gels were cast 24 hours before use following the protocol presented by Schägger, 2006. Separating gels (8%) were prepared as follows: 2.5 ml of AB-3 (49.5% T, 3% C), 2.5 ml gel buffer (3 M Tris, 1 M HCl, 0.3% SDS, pH 8.45), 0.75 ml glycerol and 9.25 ml milliQ. Polymerisation was initiated by the addition of 90 μl of 10% APS and 9 μl of TEMED. Stacking gels were prepared at a concentration of 4% acrylamide (0.5 ml AB-3, 1.5 ml 3 x gel buffer, 4 ml milliQ, 45 μl 10% APS and 4.5 μl TEMED). Gels were cast and electrophoresed using a BioRad Mini-PROTEAN® electrophoresis system. Anode and cathode buffers were prepared according to Schägger, 2006.

**In-gel digestion with trypsin.**   Removal of Coomassie G-250 was performed by incubating gel pieces for 15 minutes with shaking at 550 rpm at 37˚C in 100 μl of 100 mM ammonium carbonate prepared with 50% acetonitrile. The solution was removed and, gel pieces were then incubated for 15 minutes with shaking at 550 rpm at 37˚C in 100 μl of 100% acetonitrile. The supernatant was removed, and gel pieces were dehydrated for 15 minutes at 40˚C using Speed-Vac. Gel pieces were rehydrated in 100 μl of 50 mM Tris-HCl at pH 8.5 for 5 minutes with shaking at 550 rpm at 37˚C. The solution was discarded, and gel pieces were washed for 5 minutes shaking at 550 rpm at 37˚C prior to discarding the solution. Reduction and alkylation of cysteine residues was facilitated by the incubation of gel pieces for 10 minutes with shaking at 550 rpm in the dark at 95˚C in 100 μl of 50 mM Tris-HCl containing 10 mM Tris(2-carboxyethyl) phosphin (TCEP) and 40 mM chloroacetamide. The supernatant was discarded, and gel pieces were washed in 100 μl of 100% acetonitrile for 5 minutes at 37˚C with shaking at 550 rpm. After removing the supernatant, the step was repeated with 100 μl of 100 mM ammonium carbonate and then with 100 μl of 100% acetonitrile. Gel pieces were then dried using Speed-Vac for 15 minutes at 40˚C. Dry gel pieces were reswelled on ice with the incremental addition of small volumes (5–10 μl) of digestion buffer (41.6 mM ammonium carbonate, 5 mM calcium chloride and 0.0125 μg/μl modified porcine trypsin from Promega (Wisconsin, USA). Gel pieces were then covered with 20 μl of incubation buffer (41.6 mM ammonium carbonate containing 5 mM calcium chloride). Thereafter, digestion was facilitated at 37˚C overnight with shaking at 550 rpm. Samples were centrifuged and mixed with 15 μl of 25 mM ammonium carbonate for 15 minutes at 37˚C. The supernatant was collected, and the step was repeated with 150 μl of acetonitrile, followed by 40 μl of 5% formic acid and lastly with 150 μl of acetonitrile. The supernatant containing peptides was collected at each step and pooled following the overnight digestion. Peptides were dried for two hours at 30˚C using a Speed-Vac.

**LC-MS/MS analysis of in-gel digested proteins.**   Dry peptide samples were dissolved and acidified in 0.1% formic acid and 5% acetonitrile. Peptides were separated by nano-HPLC using a Dionex Ultimate 3000 equipped with an enrichment column (C18, 5 μm, 100 Å, 5 x 0.3 mm) and an Acclaim PepMap RSLC nanocolumn (C18, 2 μm, 100 Å, 500 x 0.075 mm) (Thermo Fisher Scientific, Vienna, Austria). Peptides were concentrated for 6 minutes at a flow rate 5 μl/min on the enrichment column using 0.1% formic acid as isocratic solvent. Peptides were separated using the nanocolumn at 60˚C with a flow rate of 250 nl/min with a gradient between 0.1% formic acid in water (A) and 0.1% formic acid and acetonitrile (B). The gradient was set up as follows: 0–6 minutes at 4% B; 6–94 minutes at 4–25% B; 94–99 minutes

at 25–95% B; 99–109 minutes at 95% B; 109.1–124 minutes at 4% B. Sample ionisation was facilitated by the nanospray source equipped with stainless steel emitters (ES528, Thermo Fisher Scientific, Vienna, Austria). Mass spectrometry analysis was performed using an Orbitrap velos pro mass spectrometer (Thermo Fisher Scientific, Massachusetts, USA) operated in positive ion mode, applying alternating full scan MS (m/z 400 to 2000) in the ion cyclotron and MS/MS by CID of the 20 most intense peaks with dynamic exclusion enabled. The LC-MS/MS data were analysed with Proteome Discoverer 1.4 (ThermoFischer Scientific) and Mascot 2.4.1 (MatrixScience, London, UK) by searching against the 6-frame translation of the *H. hemerocallidea* transcriptome assembled here as well as all common contaminants. Cysteine carbamidomethylation was set as fixed and methionine oxidation was set as variable modification. Detailed search criteria were used as follows: semitrypsin; max. missed cleavage sites: 2; search mode: MS/MS ion search with decoy database search included; precursor mass tolerance +/- 10 ppm; product mass tolerance +/- 0.7 Da; acceptance parameters: 1% false discovery rate (FDR); only rank 1 peptides; minimum Mascot ion score 20; minimum 2 peptides per protein.

## Results

### Decontamination of transcripts reminiscent from multiplex sequencing

RNA extracted from the corm, leaf and flower of *H. hemerocallidea* was sequenced on the Illumina Hi-Seq 2500 platform to generate the first *de novo* transcriptome of the phytomedicinal plant. After trimming the raw reads with Trimmomatic, more than 97% of the reads had a Phred score larger than 30 where base read accuracy is at 99.9% (S1 Fig). A total of 35,087,914 trimmed fragments (S2 Fig) were then assembled *de novo* with Trinity. From the assembled transcripts, 74,652 contaminating transcript sequences associated with *Helianthus annuus* (reminiscent from multiplex sample sequencing) were removed using DeconSeq and 143,549 clean transcripts were retained as part of the assembly for downstream analysis. A blastn search of the transcriptome assembled here against the transcriptomes of the top similar species as well as *H. annuus* (S1 Table) was used to produce a sequence similarity profile to evaluate the level of contamination before and after the decontamination step performed with DeconSeq (S3A and S3B Fig). The sequence similarity profile exhibited by *H. annuus* is hyperbolic, peaking at 100% identity with 24,712 sequences. On the other hand, the profiles of the top similar species such as *Elaeis guineensis* and *Phoenix dactylifera* exhibit a bell-shaped curve, peaking between 80% and 90% sequence similarity.

### Assembly quality and completeness

The length range of assembled transcripts was from 201 bp to 5,874 bp with an N50 length of 409 bp and a mean length of 389 bp. Transcript length distribution is depicted in S4 Fig. The completeness of the transcriptome was assessed with BUSCO based on the Liliopsida class which revealed that the transcriptome contains 21.7% complete genes (10.4% of which were single transcripts whereas 11.3% where duplicates) and 20.7% fragmented transcripts. Altogether, BUSCO accounted for 42.4% of the benchmarked orthologs expected while 57.6% were declared missing after searching 3,278 BUSCO groups (Table 1).

### Functional annotation overview

From 143,549 transcripts assembled, 68,166 (47.5%) were annotated with an e-value cut-off of $1e^{-5}$ on the COG, GO, KEGG, nr, pfam and Swiss-Prot databases. A vast majority of the transcripts annotated by the different databases were included in the nr database, itself accounting for 66,604 transcripts (46.4%). The GO and pfam databases provided annotation to 53,330 and 24,668

**Table 1. Statistical summary of the Trinity de novo transcriptome assembly of *Hypoxis hemerocallidea*.**

| | |
|---|---|
| Total transcripts | 143,549 |
| Total bases | 55,859,534 |
| **Length Distribution** | |
| Mean sequence length | 389 bp |
| Length range interval | 201 bp– 5874 bp |
| **GC content** | |
| Mean GC content | 41.95% |
| **Assembly quality measure** | |
| N50 length | 409 bp |
| **Transcriptome completeness assessment** | |
| BUSCO Liliopsida | C:21.7%[S:10.4%,D:11.3%],F:20.7%,M:57.6%,n:3278 |

transcripts respectively, both of which were entirely framed within the nr database. With standard settings, TransDecoder identifies ORFs at least 100 amino acids long. Since 80,726 (56.2%) of the transcripts have a length between 200 and 299 nucleotides long, they were discarded prior to ORF prediction. From the remaining 62,823 transcripts longer than 299 nucleotides, 55,474ORFs encoded by 38,167 transcripts were identified, of which 2,845 transcripts were not annotated on any of the databases. In that same regard, 71% of the transcripts shorter than 300 nucleotides could not be annotated on the COG, KEGG, nr and Swiss-Prot databases (S4 Fig). The overlap of annotated transcripts between the COG, KEGG, nr and Swiss-Prot databases and the ORFs is displayed in a proportional manner in the Euler diagram in S5 Fig. The Swiss-Prot database has also provided annotation to 51 transcripts that the other databases did not, as presented in the Venn diagram (S5A Fig) which correlates with the Euler diagram in S5B Fig.

## Taxonomic distribution of annotated transcripts

Taxonomic distribution analysis of annotated transcripts performed with FunctionAnnotator [33] identified the species in which most transcripts matched. The two most similar species were *Elaeis guineensis* (African oil palm) and *Phoenix dactylifera* (date palm), each identifying around 17,000 transcripts. In third place, 7,341 transcripts were identified based on *Musa acuminata* (banana). The remaining annotated transcripts (19,340) were assigned between 1,451 other species (S6 Fig).

*H. hemerocallidea* is classified under the Asparagales order in the Hypoxidaceae family. A taxonomic common tree with the top similar species as well as three members from the Asparagales order is presented in S7 Fig. A blastn profile of the transcriptome assembled here against transcriptomic assemblies of the top similar species as well as *Asparagus officinalis* (garden asparagus), *Dendrobium catenatum* and *Phalaenopsis equestris* (belonging to the Asparagales order) was carried out with a 1e$^{-20}$ e-value cut-off. The blastn search against *A. officinalis* exhibits a similarity profile to *M. acuminata* and accounts for a comparable number of matches, 10,364 and 11,197 respectively. Whilst blastn search profiles of *E. guineensis* and *P. dactylifera* yielded 15,129 and 15,066 matches respectively (S8 Fig).

## Gene ontology (GO) annotation and enrichment

Blast2GO annotated 53,330 transcripts with 6,236 unique GO terms on 366,130 instances in the top-level categories: cellular component (CC), molecular function (MF) and biological processes (BP). The most abundant groups identified in cellular components were 'cell part', 'organelle' and 'membrane'. 'Binding' and 'catalytic activity' were the predominant groups detected in molecular functions. These represented a large proportion in comparison to the third most abundant MF

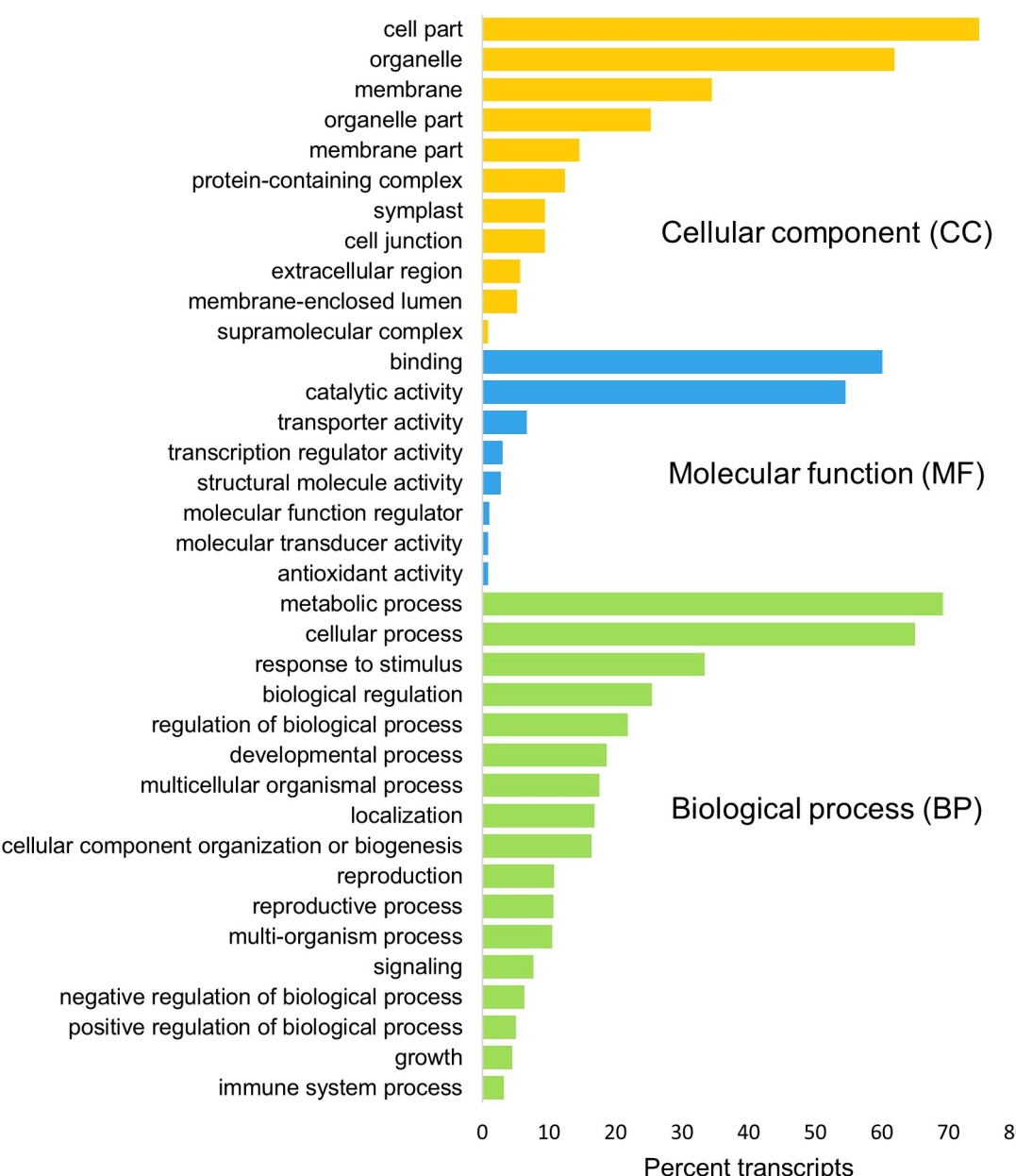

**Fig 1. Gene ontology (GO) level 2 classification and proportion of assembled transcripts from *Hypoxis hemerocallidea*.** Classification is grouped under the GO domains: 'cellular component', 'molecular function' and 'biological process'.

group detected–'transporter activity'. The top three biological processes observed were grouped under 'metabolic process', 'cellular process' and 'response to stimulus' (Fig 1).

## Clusters of orthologous groups (COG) annotation

Annotation of 59,502 transcripts on the COG database was performed with eggNOG mapper. A substantial portion of the COG annotated transcripts (14,967) were identified to have orthology to Cluster S (Function unknown). Many transcripts annotated were also clustered under groups related to protein translation, modification and turnover (O and J clusters).

Transcription (K) is also a process which was significantly represented by the annotated transcripts. In the secondary metabolites biosynthesis, transport and catabolism (Q), 1,734 transcripts were clustered by orthology (S9 Fig).

### Transcription factors

The Plant Transcription Factor Database identified 656 transcription factors distributed amongst 45 transcription factor families within the transcriptome of *H. hemerocallidea*. The most abundant transcription factor families identified were MYB-related, NAC and WRKY. The least abundant transcription factors identified were BBR-BPC, BES1, CPP, NF-YB, LSD and TCP (S10 Fig).

### Enzyme classes and Pfam domains

A total of 2,281 enzymes were identified within the enzyme commission (EC) classes. Majority of enzymes identified by EC number were transferases, hydrolases and oxidoreductases. Ligases, isomerases and lyases were also identified within the transcriptome, although in lower numbers (S11 Fig). Domain annotation on the Pfam database identified 7,559 different domains occurring on 100,274 instances in 48,653 of the assembled transcripts. The two most abundant domains identified were protein kinase domain and protein tyrosine kinase with 1,251 and 1,019 hits respectively.

### Kyoto Encyclopaedia of Genes and Genomes (KEGG) pathway annotation

KEGG Pathway annotation identified 29,225 transcripts involved in various pathways. Of those, 72% were found to be involved in pathways related to metabolism. A much smaller portion (19%) of transcripts were found to be involved in genetic information processing (Fig 2A). Carbohydrate metabolism pathways are the most abundant of metabolic pathways identified accounting for 3,898 transcripts. The metabolism of lipids, amino acids, energy and nucleotides

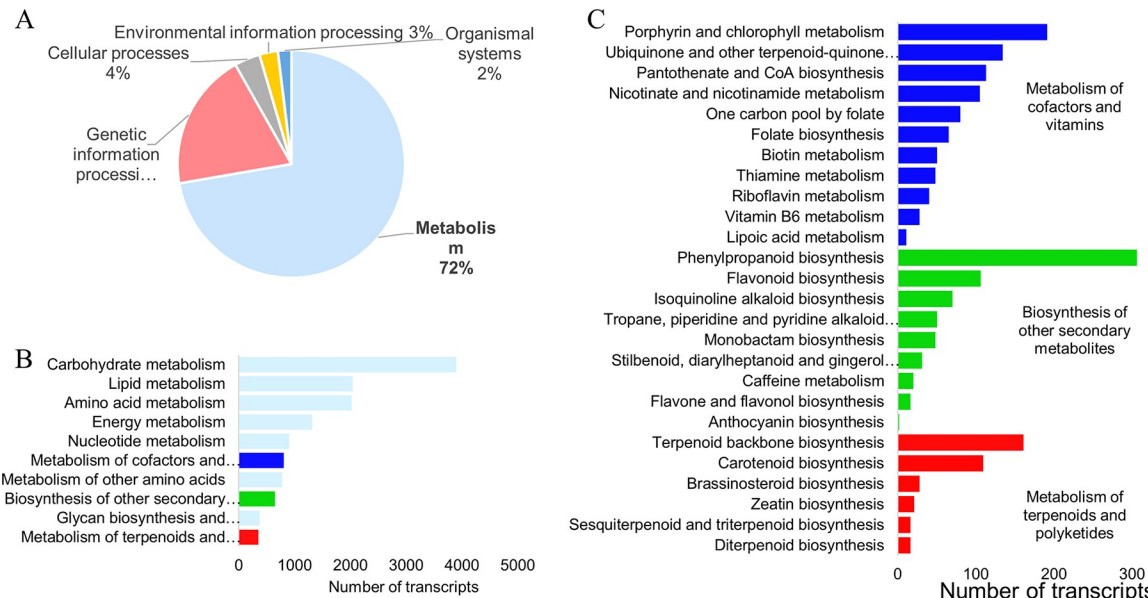

**Fig 2. KEGG pathway annotation.** A) General representation of the annotated transcripts. B) Pathway groups integrated into 'metabolism' C) Pathways grouped under three categories of metabolism': 'Metabolism of cofactors and vitamins', 'Biosynthesis of other secondary metabolites' and, 'Metabolism of terpenoids an polyketides'.

is also highly represented in the transcriptome. Transcripts were also identified in the metabolism of cofactors and vitamins (804), biosynthesis of secondary metabolites (649) and metabolism of terpenoids and polyketides (351) (Fig 2B).

The most abundant transcripts related to the metabolism of cofactors are those involved in the production of porphyrin and chlorophyll, as well as ubiquinone and terpenoid-quinones. The metabolism of B vitamins such as folate (vitamin B9), biotin (vitamin B7), thiamine (vitamin B1), riboflavin (vitamin B2) and pyridoxine (vitamin B6) are also significantly represented (Fig 2).

The biosynthesis of other secondary metabolites involved the biosynthesis of phenylpropanoids (306 transcripts), followed distantly by flavonoids (106 transcripts). The biosynthesis of isoquinoline, tropane, piperidine and pyridine alkaloids was detected as well. The biosynthesis of monobactam, a Gram-negative antibiotic, was also identified. In the stilbenoid, diarylheptanoid and gingerol biosynthesis pathway, 31 transcript isoforms were identified (Fig 2C).

The metabolism of terpenoids and polyketides is largely represented by the terpenoid backbone biosynthesis pathway and carotenoid biosynthesis. However, several transcripts were also identified to be involved in brassinosteroid biosynthesis, zeatin biosynthesis, sesquiterpenoid and triterpenoid biosynthesis and, diterpenoid biosynthesis (Fig 2C).

In the sesquiterpenoid and triterpenoid biosynthesis pathway, 10 squalene synthase isoforms were identified and 5 squalene monooxygenase transcripts. In addition, a single isoform of vestitone synthase was annotated.

Four isoforms of ent-kaur-16-ene synthase alongside two ent-kaurene oxidase isoforms and three ent-kaurenoic acid oxidase isoforms were grouped under the diterpenoid biosynthesis pathway. In the same pathway, four isoforms of gibberellin 2-beta-dioxygenase were identified.

## Differential transcript expression

Differential gene expression analysis identified a total of 946 differentially expressed transcripts, corresponding to 687 genes between the corm, leaf and flower tissues of *H. hemerocallidea*. The 946 transcripts were divided into 3 clusters, correlated by log2(FPKM+1) transformation, each being representative of genes upregulated in the leaf (1st cluster– 823 transcripts), upregulated in the corm (2nd cluster– 34 transcripts) and, upregulated in the flower (3rd cluster– 89 transcripts) (Fig 3). Transcripts upregulated in the leaf and corm are up-regulated by a mean of ~2.2 log2(FPKM+1), whereas the 89 clustered transcripts in the flower tissue are upregulated by a mean of ~4.0 log2(FPKM+1). The data and annotation of the heatmap entries can be found in the S1 File.

## Proteomic profiling of *H. hemerocallidea*

Proteomic sequencing using LC-MS/MS was performed to identify what is present in the corm, leaf and flower tissues of *H. hemerocallidea* at the proteomic level. Proteomic profiling of the three tissues has identified a total of 3,927 proteins corresponding to 3,805 transcripts which further correspond to 1,577 genes assembled here (S12 Fig). The flower tissue contained the largest number of proteins (2,813) which mapped to 2,746 transcripts followed closely by the leaf tissue with 2,636 proteins that were mapped to 2,567 transcripts. Lastly the corm tissue contained 573 proteins mapping to 550 transcripts (S12 Fig).

A significantly larger proportion of proteins were mapped back to the transcriptomic assembly when extracted in SDS as opposed to soluble extraction conditions in P-PER from ThermoScientific®. Nevertheless, some proteins undetected in SDS extraction were detected in a soluble state in P-PER (Fig 4 and S13 Fig). Moreover, fractionation of soluble corm

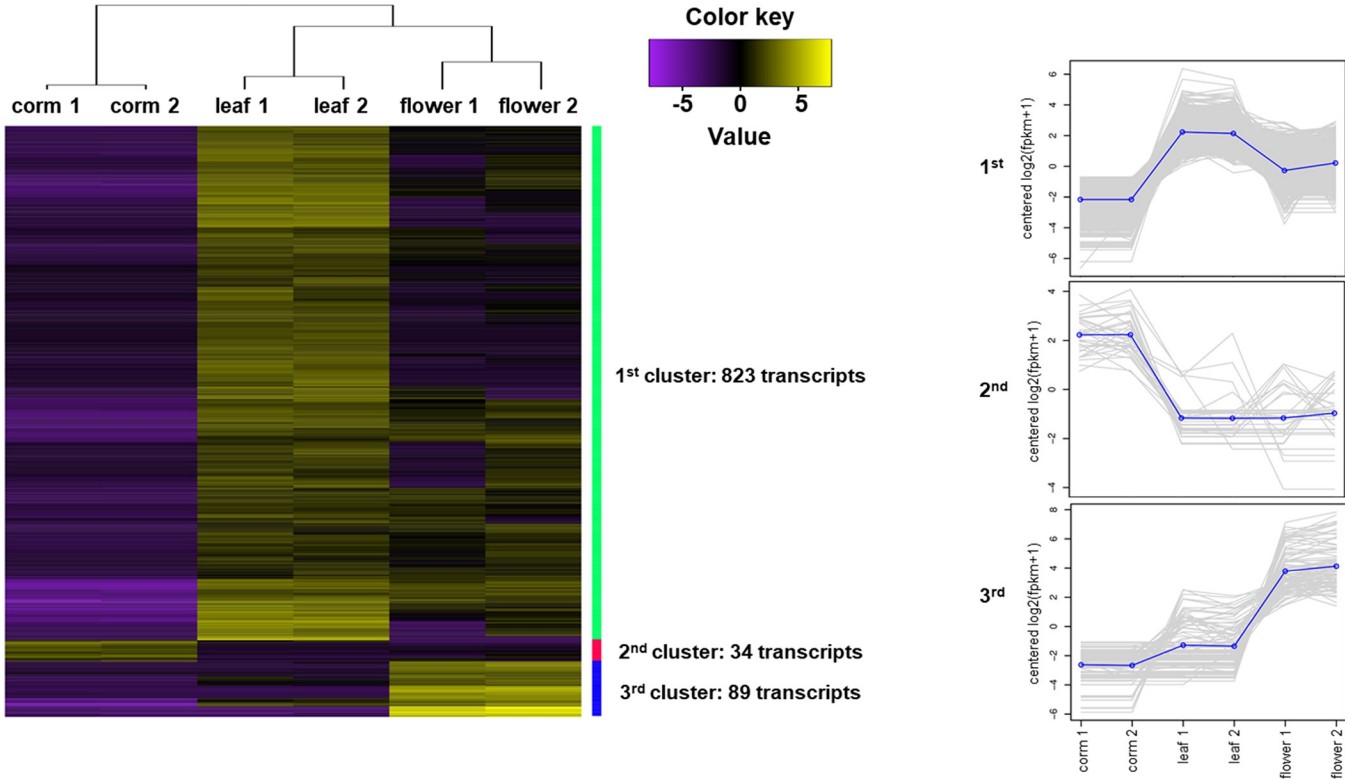

**Fig 3. Heatmap of the differentially expressed transcripts between the tuber, leaf and flower tissues of *H. hemerocallidea*.** The heatmap was generated with a p-value of 0.05 and a one-fold change cut-off for the false discovery rate of the differential expression analysis performed with edgeR. The 1st cluster of 823 transcripts was found to be upregulated in the leaf tissue with a mean log2(fpkm+1) of 2.2. A similar mean is exhibited by the 34 transcripts upregulated in the corm tissue (2nd cluster). However, the mean log2(fpkm+1) of the upregulated transcripts in the flower tissue (3rd cluster) is double that of the tuber and leaf upregulated transcripts. Line plots depict the expression levels of transcripts for each of the three clusters.

proteins assisted in the identification of an additional 118 proteins (Fig 4C) which were not previously identified in the P-PER or SDS proteomic extraction methods in the corm tissue. While across all tissues, fractionation assisted in the identification of 61 proteins that are exclusive to the method (S13 Fig).

From the 946 differentially expressed transcripts presented in Fig 3, a total of 365 upregulated transcripts have been confirmed proteomically across all tissues (Fig 4). Tissue specifically, 291 of the 823 upregulated transcripts were confirmed in the leaf tissue by proteins extracted from the leaf; 33 of 89 upregulated transcripts in the flower tissue were confirmed by proteins extracted from the flower and 4 of 34 upregulated transcripts were confirmed proteomically within the corm tissue. A total remainder of 37 upregulated transcripts were confirmed proteomically, however, in a different tissue than that in which the transcript was upregulated in.

## Overview of secondary metabolism

Secondary metabolism pathways were analysed at three levels of information. Firstly, by the presence of transcripts involved in secondary metabolism annotated in the *H. hemerocallidea* transcriptome. Secondly, through differentially expressed transcripts between the corm, leaf and flower tissues. Thirdly, through the qualitative detection of proteins by LC-MS/MS (Fig 5). At the level of the assembled transcriptome, transcripts were found in the terpenoid backbone biosynthesis via the mevalonate pathway and the methylerythriol pathway (MEP) as

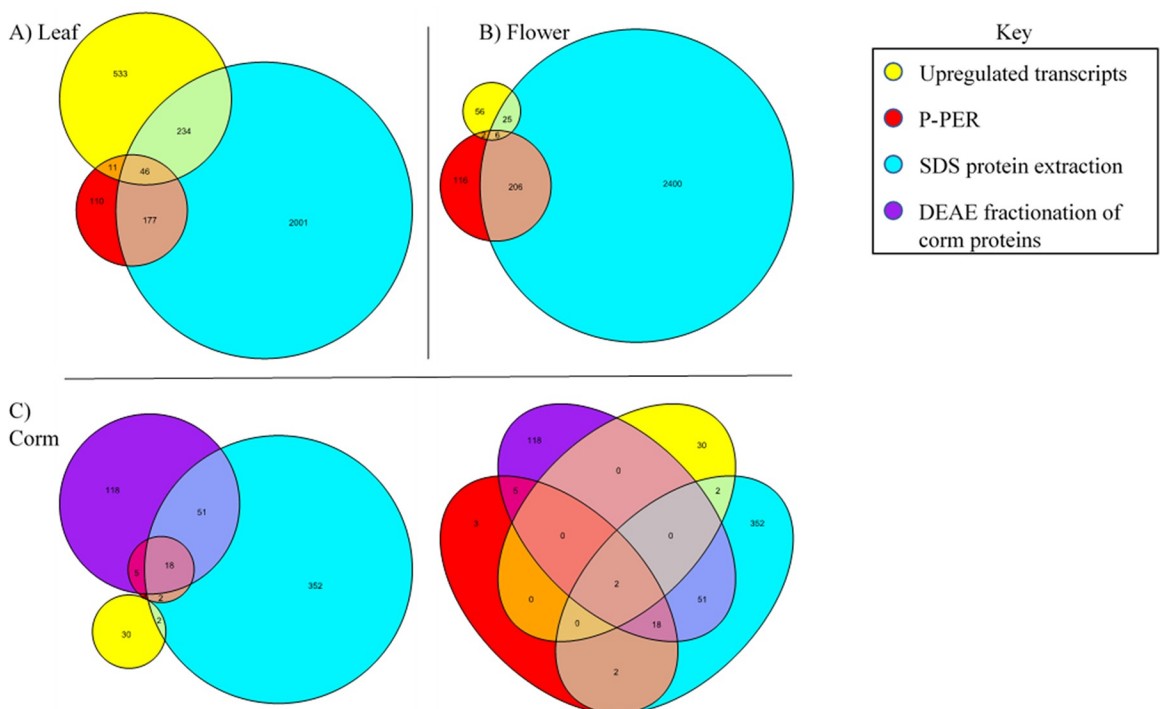

**Fig 4. Proteomic confirmation of upregulated transcripts in the leaf, flower and corm tissues.** The proteomic extraction methods are exhibited proportionally in Euler diagrams for A) leaf, B) Flower and C) Corm. A Venn diagram is included in C to display the 2 transcripts identified in all proteomic extracts as well as upregulated transcriptomically.

well as in the production of carotenes, alpha-carotene, xanthophylls, apocarotenoids, cycloartenol and terpenes. In the biosynthesis of phenolics, transcripts were identified in the production of chalcones, p-coumaroyl-CoA, flavonones, aureones, dihydroflavonols, anthocyanidins, flavonols and flavonol glycosides. Several transcripts were also identified in the production of glucosinolates as well as in the biosynthesis of alkaloids. Upregulated transcripts and expressed proteins were also identified in the biosynthesis of terpenoids and phenolics primarily within the leaf tissue, followed by the flower tissue.

The leaf tissue was identified to be the most active of the tissues in the secondary metabolite pathway; it was found to upregulate transcripts in the MEP pathway, as well as in the production of carotenes, alpha-carotene, xanthophylls, apocarotenoids and terpenes. Regarding phenolics, upregulated transcripts were identified in the formation of chalcones and anthocyanidins (Fig 5). At the proteomic level, the leaf tissue expresses proteins in mevalonate pathway and the MEP pathway. Further, the production of carotenes, xanthophylls and apocarotenoids was proteomically confirmed in the leaf tissue alongside phenolics in the production of p-coumaroyl-CoA, chalcones, aureones, flavones and, flavonol glycosides (Fig 5).

In the flower tissue, the production of alpha-carotene is higher than in the leaf, however, not in both replicates of the flower tissue. On the other hand, the production of aureones is significantly elevated in the flower tissue compared to the leaf and corm (Fig 5). At the proteomic level, the flower tissue expresses proteins in mevalonate pathway and the MEP pathway and in the production of isopentenyl pyrophosphate. Further, the production of carotenes and apocarotenoids was proteomically confirmed in the flower tissue. Regarding phenolics, the production of p-coumaroyl-CoA, chalcones, aureones, flavones, flavonols and glycosides are predominantly found within the leaf and flower tissues (Fig 5).

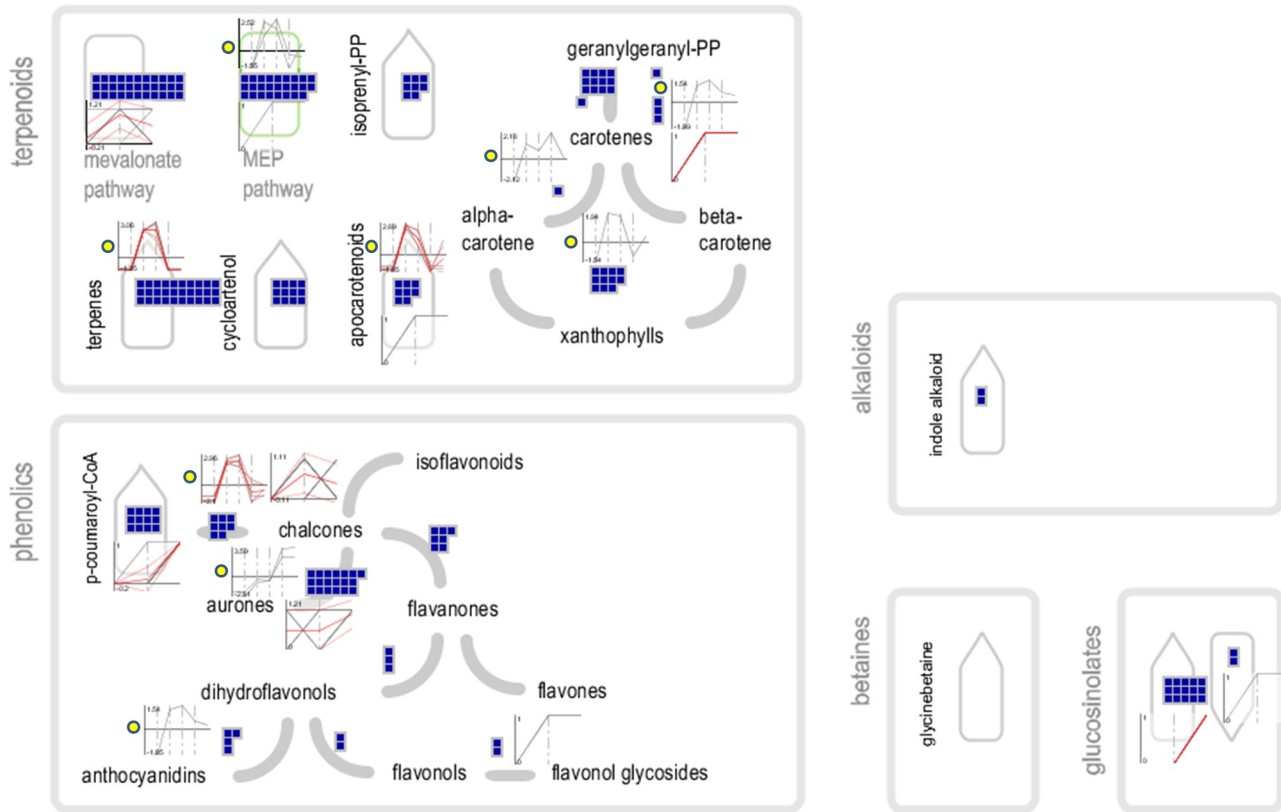

**Fig 5. Overview of secondary metabolism.** Blue squares indicate assembled transcripts. The 6-point line graphs with a yellow circle next to them are plots of differentially expressed transcripts in duplicate for the corm, leaf and flower tissues with $\log_2$(FPKM +1) on the vertical axis. The 3-point line graphs that are not otherwise marked, represent the qualitative detection of proteins in the corm, leaf and flower tissues.

There were no upregulated transcripts involved in secondary metabolism that were identified in the corm tissue. However, corm proteins were detected in the mevalonate pathway and in the production of aureones, namely four aureusidin synthase proteins isoforms (Fig 5).

## Terpene biosynthesis

In the terpenoid biosynthesis pathway, transcripts were annotated by Mercator4 and their annotation corresponds to that from the nr database. Transcripts were identified at every step of the mevalonate pathway (MVP) as well as of the methylerithriol pathway (MEP). Although, geranyl pyrophosphate synthase was not identified, 4 linalool synthase transcripts and 6 α-terpineol synthase transcripts, both monoterpenoid synthases which use geranyl pyrophosphate as a substrate, were identified. Three of the 4 linalool synthase transcripts were found to be upregulated in the leaf tissue–and those were the only terpenoid synthases found to be upregulated transcriptomically. Similarly, there were no terpenoid synthases detected at the proteomic level (Fig 6).

Regarding the synthesis of sesquiterpenoids, 4 farnesyl pyrophosphate synthase transcripts were annotated, the enzymes which produce substrates for sesquiterpenoid biosynthesis. Transcripts were found to produce four different sesquiterpenoid synthase. More specifically, two transcripts of nerolidol synthase, and one transcript of each α-humulene synthase, germacarene-D synthase and valencene synthase (Fig 6).

Four geranylgeranyl pyrophosphate synthase transcripts were identified, the enzymes which produce substrates for diterpenoid biosynthesis. There were no diterpenoid synthases

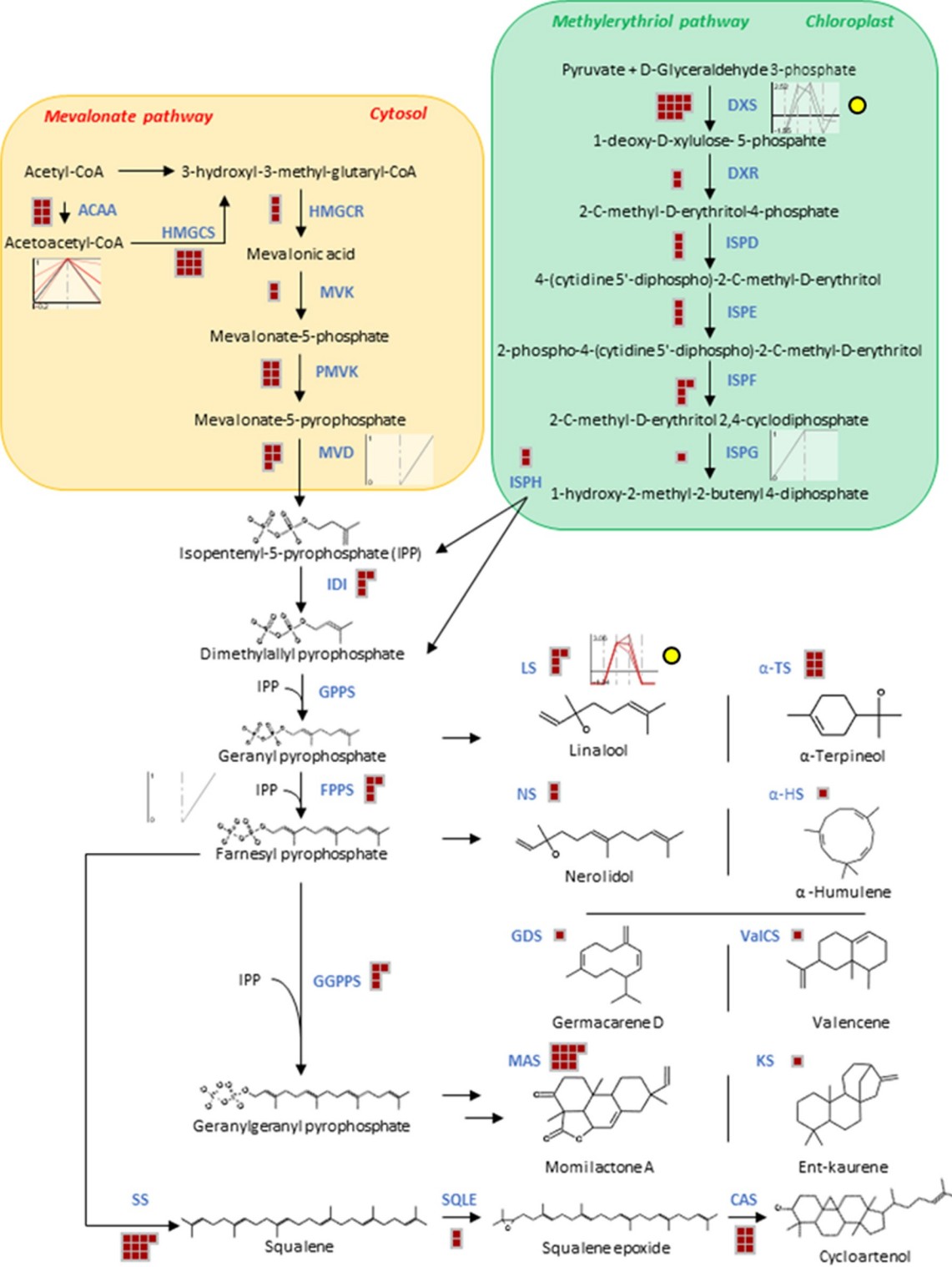

**Fig 6. Terpene biosynthesis.** Red squares indicate assembled transcripts. The 6-point line graphs with a yellow circle next to them are plots of differentially expressed transcripts in duplicate for the corm, leaf and flower tissues with $\log_2(\text{FPKM} +1)$ on the vertical axis. The 3-point line graphs that are not otherwise marked, represent the qualitative detection of proteins in the corm, leaf and flower tissues.

annotated by Mercator4. However, 10 transcripts were annotated as momilactone A synthase and one transcript was annotated as an ent-16-kaurene synthase. Both of those directly involved in the synthesis of their respective diterpenoid phytoalexins, using enzymatically transformed geranylgeranyl pyrophosphate (Fig 6).

Six transcripts of the triterpenoid synthase, cycloartenol synthase, were annotated by Mercator4; while 10 squalene synthase transcripts and 2 squalene epoxidase transcripts were identified which provide the squalene epoxide substrate for the synthesis of cycloartenol (Fig 6).

Differential expression analysis revealed that two 1-deoxy-D-xylulose 5-phosphate synthase (DXS) transcripts were upregulated in the MEP pathway within the leaf although transcripts were detected at every step of the MEP pathway. At the proteomic level, in the MEP pathway, only a 4-hydroxy-3-methylbut-2-enyl diphosphate synthase (ISPG) isoform was detected in the leaf and flower. In the mevalonate pathway there were no transcripts found to be upregulated between the three tissue types. However, 5s acetyl-CoA C-acyltransferase proteins were identified within the leaf. One of those was identified in all three tissues and a second one was identified in the leaf and flower only. Also, in the mevalonate pathway, a mevalonate diphosphate decarboxylase (MVD) protein isoform was identified in the flower tissue alone (Fig 6).

Additional annotation of terpenoid synthases was obtained by gene ontology annotation in the molecular function category. Aside from the terpene synthases annotated on the nr database and by Mercator4, transcripts were annotated to possess activity as various terpene synthases. One transcript was annotated by gene ontology to be active as a germacrene-D synthase–this transcript was found to be upregulated (S1 File) in the leaf alongside the three upregulated linalool synthase isoforms. Further, three terpene synthase10-like isoforms without a defined terpenoid synthase function by the nr database, were annotated by gene ontology to have myrcene synthase, (E)-beta-ocimene synthase and (R)-limonene synthase activity. Moreover, the multiple terpene synthase activity is attributed to a single nr-annotated terpene synthase by GO annotation. More specifically, cycloartenol synthases annotated by nr, were associated with arabidiol, marneral, beta-amyrin, baruol, thalianol, cycloartenol and lupeol synthase activity by GO annotation. Regarding lupeol, additional transcripts were attributed with GO lupeol synthase activity like pleiotropic drug resistance protein 12-like isoform X1transcripts and dual-specificity protein phosphatase PTEN.

The nr-annotated valencene synthases were attributed epi-cedrol synthase activity by GO annotation. The nr-annotated linalool and alpha-terpineol synthases were both attributed with sabinene, alpha-farnesene, (R)-limonene, (4S)-limonene, pinene and (E)-beta-ocimene. The nr-annotated alpha humulene, was attributed with (+)-delta-cadinene synthase activity by GO annotation. There also nr-annotated terpene synthases that were not annotated by Mercator4 (thus not shown in Fig 6) that were correspondingly annotated by GO like S-(+)-linalool synthase and (-)-E-beta-caryophyllene (S14 Fig).

## Discussion

### Decontamination

The decontamination profiles (S3 Fig) indicate that DeconSeq [29] has identified the 24,712 sequences belonging to *H. annuus* with 100% identity as well as the remaining 49,940 sequences with lower than 100% sequence identity which are likely misassembled sequences caused by the homologous sequences present between both species. There is a hyperbolic decrease in the number of sequences matching the transcriptome of *H. annuus* as the percentage identity drops. This would not come as a surprise normally, however, the sequence similarity profiles of the top species, differ significantly in that they are bell-curve shaped and not hyperbolic. More in contrast, the peak of top similar species is several percentiles below 90%.

The sequence similarity profiles suggest that contaminant assemblies produce a hyperbolic profile when blastn searched against their respective genome/transcriptome. Sequence similarity profiles like the ones produced here could be used to assess the efficacy of decontamination procedures in cases where -omic data is available for the contaminating sequences. Overall, DeconSeq has proven to remove contaminating sequences precisely. Notably, considering that DeconSeq identified contaminants below the percentage identity threshold of 95% recommended for the removal of contaminants by newer tools such as CroCo [45]. While in this case, several thousand contaminating sequences were present below 95% sequence identity.

## Assembly quality and completeness

The N50 length of 409 bp (Table 1) the contig length distribution (S4 Fig) of the assembled contigs are comparable to some of the plant transcriptomic assemblies available with an N50 ranging from 168 bp to 535 bp [46, 47]. In retrospect, a large N50 can sometimes be caused by the miss-assembly of long chimeras [48]. Nonetheless, BUSCO has accounted for 42.4% of genes in the Liliopsida class, 20.7% of which were incomplete fragments indicating that there is significant room left for expanding on the transcriptome of *Hypoxis hemerocallidea* assembled here. This is not surprising given that the mature *H. hemerocallidea* plant material used in this study may be lowly active at a transcriptomic level due to tissue dormancy. Especially regarding the corm, a storage tissue which may exhibit comparable reduced transcriptomic activity to the tuber of *Solanum tuberosum* [49]. This is also indicated by the relatively low number of reads obtained from the corm tissue compared to the leaf and flower tissues (S2 Fig). In addition, there is no evidence regarding the number of chromosomes within *H. hemerocallidea* or the variety of transcripts produced.

## Functional annotation overview

Transcript annotation was achieved for 47.5% of the 143,549 transcripts across the COG, GO, KEGG Pathway, nr, pfam and Swiss-Prot databases. A large portion of the un-annotated transcripts (57,287 or, 39.9%) were sequences shorter than 300 nucleotides long (S4 Fig) which were not annotated likely due to the higher e-values associated with the short sequence length [50]. Annotation based on manually curated databases such as UniProtKB/Swiss-Prot [51] and KEGG [52, 53] yielded lower numbers of annotated transcripts compared to the nr and COG databases which include predicted proteins as well as manually curated. In addition, it is no surprise that annotation on the KEGG database identified the least number of sequences since KEGG annotation comprises of manually curated sequences with a known involvement in a pathway and sequences annotated with an unknown function are excluded. In that, the largest cluster of genes annotated on the COG database identifies 14,967 transcripts with an unknown function [S] (S9 Fig). The large number of transcripts with an unknown function indicates the gap in knowledge regarding the omics of the *Hypoxis* genus. Further adding to that gap, are the 2,845 ORF encoding transcripts identified using Transdecoder which were not annotated across any of the databases used.

## Taxonomic distribution of annotated transcripts

*Hypoxis hemerocallidea* is classified under the Asparagales order and it bares bulk transcriptomic similarity to *Asparagus officinalis* which belongs to the same order. The transcriptome assembled here also contains a larger number of transcripts matching with high similarity to members of the order Arecales (Arecaceae) than Asparagales. Numerous transcripts matched closely to transcripts from *Musa acuminata* (banana) belonging to the Zingiberales order (S6 Fig). The number of matches to either species may reflect the

expression and processing of the transcriptome of the *H. hemerocallidea* produced here and not necessarily the accuracy of the classification of the organism (S7 Fig). Although, this study does highlight that there are numerous similarities that the transcriptome of *H. hemerocallidea* shares with the transcriptomes of *E. guineensis* and *P. dactylifera*, and about two thirds as much with *M. acuminata* and *A. officinalis* (S8 Fig). It is perhaps due to a higher similarity to the Arecales order than to Asparagales order that *A. officinalis* or another species from the Asparagales order was not identified as one of the top similar species by FunctionAnnotator. In addition, 29% of the annotated transcripts were annotated between a broad list of 1,451 species (S6 Fig) which may indicate the lack of sequences available for organisms closely related to *Hypoxis* and suggests the presence of a versatile set of genes within *H. hemerocallidea*.

## Differential expression

The largest cluster of upregulated transcripts comparing the corm, leaf and flower tissues of *H. hemerocallidea* was found within the leaf tissues while the highest level of upregulation was found within the flower tissue. The corm tissue had the lowest number of transcripts upregulated (Fig 3). The data was normalised for the respective sample sizes of the tissues by using FPKM (fragments per kilobase million) and further transformed logarithmically. Thus, the differences in transcript expression may be reflective of the transcripts prioritised by each of the tissue types at least at the time of harvest of the two-year-old plant. Likewise, the low number of reads obtained from the corm tissue comparative to the other two tissues (S3 Fig) may indicate that the tissue has slowed down the production of metabolites and is facilitating storage. In contrast, the flower tissue which is the newest part of the plant and is a developing seasonal reproductive tissue, it is not surprising that it has the highest level of upregulation for some transcripts even though the number of reads from the flower tissue is significantly lower than that of the leaf.

## Proteomic profiling

Like the RNA-seq results, the corm tissue yielded the fewest unique proteins comparative to the leaf and flower. In contrast to the RNA-seq results, the flower tissue yielded the largest number of unique proteins compared to the leaf and flower. Together with the differential transcript expression analysis, this indicates that although the leaf upregulates a diverse cluster of transcripts with various functions, the flower tissue is significantly more active in the production of some transcripts and proteins. This phenomenon is precedented as a higher number of proteins was reported in the flower than in the leaf of Cowpea [54]. The number of identified proteins is much lower than that of assembled transcripts. Having identified 3,927 unique proteins and 143,549 transcripts across the three tissues, this producing a proportion of one protein to every ~36.6 transcripts (or 2.7%) at a global level. However, this ratio is much higher when comparing the number of proteins identified within the upregulated transcripts with the number of upregulated transcripts. That is, 328 protein matches out of 956 upregulated transcripts resulting in one protein for every ~2.9 upregulated transcripts (or 34.7%) (Figs 3 and 4). This indicates that the proteomic profiling is reflective of the upregulated transcripts identified within the 3 tissues and serves to validate to a large extent the differentially expressed transcripts identified in Fig 4.

## Terpenoid biosynthesis

Both the MVP and the MEP pathways for the biosynthesis of isoprenoid precursors have been confirmed in the transcriptome of *H. hemerocallidea*. Moreover, aside from geranyl

pyrophosphate synthase, the synthases known to produce the respective sesquiterpenoid, diterpenoid and triterpenoid precursors have been identified as well. It is likely that the geranyl pyrophosphate synthase is also present but may not have been present in enough abundance to be sequenced. From the numerous terpenoid synthases identified transcriptomically, only linalool synthase was found to be upregulated and only in the leaf tissue. This indicates that the terpenoid profile of the three tissue is not significantly upregulated to highlight the actual differences except for a minor few.

There were no terpene synthase enzymes detected in a soluble or insoluble state which indicates two possible causalities. Either very low terpene synthase protein expression was present–undetectable by LC-MS/MS. Or, the protein extraction methods applied here were not efficient at extracting the terpene synthases present.

The various nr-annotated terpene synthases attributed with multiple GO terpene synthase activity is noteworthy (S14 Fig). However, that is plausible since the concept of multi product terpene synthases was confirmed in terpene synthases from sandalwood [55]. Nevertheless, there is some level of certainty associated with the terpenoid synthases annotated here. Several of the products of the terpene synthases annotated here have been detected by GC and GC-MS in the leaf and corm tissue. Specifically, sabinene, linalool, α-terpineol, β-caryophyllene, cis-nerolidol, myrcene, trans-β-ocimene, δ-cadinene, limonene. Additionally, α-caryophyllene was identified [19], however, beta-caryophyllene synthase was identified in the transcriptome assembled here.

## Conclusion

The transcriptome and proteome of the corm, leaf and flower tissues of *Hypoxis hemerocallidea* were successfully sequenced. A 47.5% annotation rate of transcripts was achieved using the COG, GO, KEGG, nr, pfam and Swiss-Prot databases. Transcriptome annotation has identified various terpene synthase transcripts. Differential expression analysis revealed that only linalool synthase is upregulated in the leaf tissue, whereas no other terpene synthase was found to be upregulated between the three tissues. This study is the first transcriptome and proteome produced for *H. hemerocallidea*, expanding the genetic resources available for this plant.

## Supporting information

**S1 Fig. FastQC mean quality scores of the raw and trimmed reads created with MultiQC.** Reads were trimmed using Trimmomatic 0.36 using parameters described in the methods and materials. After trimming, more than 97% of the reads have a phred score above 30. The colour of the lines depicts the quality of the reads. Green indicates good quality; orange indicates medium quality and red indicates low quality.
(TIF)

**S2 Fig. Number of trimmed fragments from the corm, leaf and flower of *H hemerocallidea*.** The total number of fragments is 35,087,914 between all sample replicates.
(TIF)

**S3 Fig. Sequence similarity profiles of the blastn search of the transcriptome of *H. hemerocallidea* assembled here to top similar species.** (A) Before removal of *Helianthus annuus* contaminating sequences originating from the multiplex experiment setup and (B) after removal of contaminating sequences with DeconSeq.
(TIF)

**S4 Fig. Contig length distribution of the 143,549 transcripts from *Hypoxis hemerocallidea* assembled here.** Transcripts annotated between the COG, KEGG, nr and Swiss-Prot databases

with a cut-off e-value of 1e$^{-5}$ are coloured in orange. A majority portion of the transcripts with a length below 300 nucleotides were not annotated (57,287 of 80,726 transcripts shorter than 300 nucleotides, i.e. 71%).
(TIF)

**S5 Fig. Transcriptome annotation using various databases.** A) Venn diagram of the assembled transcripts annotated with an e-value cut-off of 1e$^{-5}$ on the COG, KEGG, nr and Swiss-Prot databases and the transcripts encoding open reading frames (ORFs). The diagram presents the overlap of transcripts annotated between the database in a numerically accurate manner. B) Euler diagram depicting the proportional overlap between the annotation of transcript isoforms on the COG, KEGG, nr and Swiss-Prot databases as well as transcripts which were identified to encode ORFs with Transdecoder.
(TIF)

**S6 Fig. Taxonomic distribution of the 66,604 transcripts annotated on the nr database.** Many of the transcripts were closely associated with *Elaeis guineensis* (African oil palm tree) and *Phoenix dactylifera* (date palm). *Musa acuminata* subsp. *Malaccensis* (banana) accounts for a significant portion of the transcripts as well. The remaining transcripts were annotated amongst 1,451 species.
(TIF)

**S7 Fig. Taxonomic common tree including *Hypoxis hemerocallidea*, the top species in terms of number of transcripts best matching on the nr database.** In addition, three members from the Asparagales order were also included. The order and family within class Liliopsida are labelled in bold. Species are italicised. Taxonomic lineages were identified on NCBI Common Tree.
(TIF)

**S8 Fig. Sequence similarity profiles of the transcriptome of *H. hemerocallidea* assembled here blastn searched against the top similar species.** *Asparagales officinalis*, *Dendrobium catenatum* and *Phalaenopsis equestris* belonging to the Asparagales order are included as well.
(TIF)

**S9 Fig. Distribution of H. hemerocallidea transcripts into clusters of orthologous groups (COG).**
(TIF)

**S10 Fig. Transcription factor families identified in the *H. hemerocallidea* transcriptome.**
(TIF)

**S11 Fig. Annotated enzymes in the *H. hemerocallidea* transcriptome grouped by enzyme commission (EC) classes.**
(TIF)

**S12 Fig. Proteomic profiling of the corm, leaf and flower tissues of *H. hemerocallidea*.** A) A total of 3,927 proteins were identified translated from 3,805 transcripts transcribed from 1,577 genes. B) The flower tissue yielded the largest number of proteins between the three tissues.
(TIF)

**S13 Fig. Overall proteomic identification by methodology used across the corm, leaf and flower tissues of *H. hemerocallidea*.** The extraction methodologies used are, SDS extraction, P-PER (ThermofischerScientific$^{®}$) and fractionation of soluble corm proteins.
(TIF)

**S14 Fig. Gene ontology molecular function annotation of transcripts with terpene synthase activity.** Yellow–terpene synthases that although annotated by nr, Mercator4 and GO, function could +only be retrieved from GO annotation. Grey–nr annotated terpene synthases attributed with multiple terpene synthase activity by GO. Brown–GO terpene synthase activity corresponding to the terpene synthases mapped by Mercator 4 and nr. Light blue–terpene synthases annotated by Go and nr but not Mecator4.
(TIF)

**S1 Table. Name of organisms and respective RefSeq assembly accessions of genomes used with DeconSeq to retain contaminants *H. hemerocallidea.***
(TIF)

**S2 Table. Organisms and their respective RefSeq assembly accessions for the transcriptomes of the top 9 similar species to the transcriptome of H. hemerocallidea.** Helianthuus anuus was included to compare and assess decontamination efficacy.
(TIF)

**S1 File.**
(XLSX)

## Author Contributions

**Conceptualization:** Ruth Birner-Grünberger, Karl Rumbold.

**Data curation:** Selisha Ann Sooklal, Thuto Ntsowe, Previn Naicker, Stoyan Stoychev, Dirk Swanevelder.

**Formal analysis:** Previn Naicker, Barbara Darnhofer, Stoyan Stoychev.

**Investigation:** Mihai-Silviu Tomescu, Selisha Ann Sooklal, Robert Archer, Dirk Swanevelder, Ruth Birner-Grünberger.

**Methodology:** Mihai-Silviu Tomescu, Selisha Ann Sooklal, Thuto Ntsowe, Robert Archer.

**Software:** Thuto Ntsowe.

**Writing – original draft:** Mihai-Silviu Tomescu.

**Writing – review & editing:** Karl Rumbold.

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
