## [Decision Letter · Decision Letter 0]

16 Oct 2020

PONE-D-20-28875

Transcriptome and proteome of the corm, leaf and flower of Hypoxis hemerocallidea (African potato)

PLOS ONE

Dear Dr. Rumbold,

Thank you for submitting your manuscript to PLOS ONE. After careful consideration, we feel that it has merit but does not fully meet PLOS ONE’s publication criteria as it currently stands. Therefore, we invite you to submit a revised version of the manuscript that addresses the points raised during the review process.

All reviewers raised some comments and concerns about this manuscript, please make your effort to respond.

We look forward to receiving your revised manuscript.

Kind regards,

Xiang Jia Min, Ph. D.

Academic Editor

PLOS ONE

Journal Requirements:

2.Thank you for stating the following in the Financial Disclosure section:

[Our gratitude goes to the Department of Science and Technology, South Africa, for the Biocatalysis Initiative. MST received a PhD bursary from the National Research Foundation, South Africa, grant number ].   

We note that one or more of the authors are employed by a commercial company: Omics Center Graz, BioTechMed and ACIB GmbH

Reviewers' comments:

Reviewer's Responses to Questions

**Comments to the Author**

1. Is the manuscript technically sound, and do the data support the conclusions?

Reviewer #1: Yes

Reviewer #2: Yes

Reviewer #3: Yes

2. Has the statistical analysis been performed appropriately and rigorously? 

Reviewer #1: No

Reviewer #2: Yes

Reviewer #3: Yes

3. Have the authors made all data underlying the findings in their manuscript fully available?

Reviewer #1: Yes

Reviewer #2: Yes

Reviewer #3: Yes

4. Is the manuscript presented in an intelligible fashion and written in standard English?

Reviewer #1: Yes

Reviewer #2: Yes

Reviewer #3: Yes

5. Review Comments to the Author

Reviewer #1: The research question is well defined, and I believe the experiment was well designed and executed properly. However, there are a few issues, as highlighted in the basic reporting section, where some information should be clarified, or extra information should be included. In particular, please be more transparent and replicates, comparisons, and use of the same/different tissue for the qPCR.

Validity of the findings

I believe this experiment to have been executed rigorously, but the manuscript needs more detail to ensure transparency about the number of plants/replicates/trials done.

1 The differential expressed genes must be validated using qRT-PCR, please added this part correctly. Because I noticed that the author only used two replicates for transcription sequencing.

2 Association analyzed must be conducted between the differential expressed genes (DEGs) and the differential expressed proteins (DEPs) identified in your study. Positive? Negative or neither? Please added them and make it clearly.

Basic Reporting

Abstract and Introduction

Overall the information presented in the abstract and introduction is relevant and interesting and does a good job at describing the background and the biological question. However, the English is not very clear in places and therefore the manuscript would benefit from proofreading and correcting in places.

For example:

1 • Line 5-7: “The metabolites of H. hemerocallidea have been identified in several studies. More recently, the terpenoids of the plant have been identified .However, the biochemical pathways and the enzymes involved in the production of metabolites have not been characterised..”

This sentence is incompletely and needed to make clearly.

Materials and methods

The level of detail is sufficient but the English needs revision.

1 •Line 84-85 “Flower and leaf material were immediately frozen in liquid nitrogen upon collection and stored at -80 ºC until use”.

Please add the information how to collected flower and leaf material? How many plants? How many days after planting and How to keep the plant? as well as how many replicates? Make all of them carefully and clearly.

2 •Line 118-120 “The high-quality paired-end reads of the flower, leaf and corm tissue of H. hemerocallidea were concatenated and assembled de novo (in the absence of a reference genome) into a single RNA-seq dataset using Trinity version 2.6.6 under default settings”

How to eliminate for mutli-duplication’s reads, how to obtain unigene, how to assemble the contigs into one transcript, you must be make it clearly.

3 In section 153 Differential transcript expression, it is not clear what comparisons have been made. Is it any comparisons were made? Also please describe the methodology in more details, particularly with respect to replicates.

please state how you did the clustering A lot more detail is needed for this part of the analysis.

Results

1. 452 Differential transcript expression

The comparison of the reads to other species is not clear – it is missing from the materials and methods section. Please clarify the reason for these species being used, how they were compared.

2.Please add the part of “Verification of the gene expression profiles of candidate DEGs by qRT-PCR” and “Statistical analysis” In methods and results part. Please denote the association analyzed between DEGs and DEPs identified in this research.

Discussion

Content in the discussion is good and is a relevant discussion of the results. However, the English is poor and therefore would benefit from proofreading.

Reviewer #2: Manuscript present transcriptome and the proteome sequencing of medicinal African potato (Hypoxis hemerocallidea). Numerous terpene synthases were identified through functional annotation. Differential expression analysis showed that which tissue upregulateslinalool synthase.

It is valuable, because combined transcriptome and proteome analyses gives a complete insight of genetics and biochemistry of this valuable medicinal plant.

There are some comments that must be addressed by authors before possible acceptance of the manuscript:

Abstract:

- Lines 2-7: This paragraph is belongs to Introduction. Just one line is abstract is enough to shows the importance of plant of interest.

- Line 14: … such as nerolidol synthase, germacrene D synthase, ….

Introduction:

- Line 26: Hypoxidaceae must be written in Italic.

- Line 31: Please change “medical conditions” with an appropriate phrase.

- Line 32: … some cancers, and HIV-AIDS.

- Line 35: Reference “(Liebenberg, 1969)” is out of the instruction of journal. Please keep the same format for references.

- Line 36: Latter research has corroborated some of the medicinal properties (Reference??).

- Line 47: please rewrite “and so have their use in the treatment of inflammation”.

- Line 51-52: please mention the reasons why “the cultivation of H. hemerocallidea is known to be problematic”??

- Line 49-52: Author mentioned that (i) synthetic production of hypoxoside is difficult, (ii) tissue culture of H. hemerocallidea produces low yields of hypoxoside rendering this method impractical, and (iii) the cultivation of H. hemerocallidea is known to be problematic. With this testimonial, what solution is provided by the authors?

- Line 60: Please give some examples of “ other plant sources”.

- Line 60: highly unlikely??!

- Line 72: …. H. hemerocallidea was …

- Introduction section needs some examples of similar researches in other medicinal plants.

Methods and materials

- Line 84: Flower and leaf materials …

There is no additional comments related to other sections of “Methods and materials” and I appreciate the authors’ efforts for complete writing and explanations of these sections.

Results

- Line 357: ORFs: Please mention the abbreviations in their first use in the text.

Discussion

- Lines 633-650: Taxonomic distribution of annotated transcripts: there is no discussion for this paragraph and it seems that it is a “Results” section not “Discussion”.

- Lines 652-665: Paragraph “Differential expression”. It also similar to “Results” section not “Discussion”.

- Lines 685-708; Paragraph “Terpenoid biosynthesis” there is no related references in this section.

Reviewer #3: The authors sequenced the transcriptome, proteome of three tissues of Hypoxis hemerocallidea (African potato), and did the assembly and functional annotation of the transcriptome, different expression analysis of the transcripts, cross-analysis of transcriptome and proteome, and analyzed the secondary metabolism.

I have the following major concerns:

1. The authors did decontamination on the contigs after the assembly. I suggest running decontamination on the fastq files before the assembly. the reads from the other species can cause assembly errors, and the contigs can contain a mix of the reads from different species. Maybe that is why the assembly is too fragmental.

2. I don't recommend the authors run assembly by default parameters on Trinity and then stopped to refine the assembly results. The assembly results are the cornerstone of the whole project, better assembly results can avoid analysis errors. 

Some minor comments:

1. the last row of table 1 is not well described, especially the last item (n: 3278)

2. for fig 3, set a title to the two plots. the plot on the right is not suitable for the line plot. because the data are independent of each other.

3. at line 324-325, the authors say "A total of 35,087,914 fragments (S2 Fig) were then assembled de novo with Trinity.", but the Fig S2 title says "number of trimmed fragments", it is a little confusing, I suggest changing it as "number fragments after trimmed".

4. There is a yellow line in the raw reads plot in Fig S1, while the others are red. A legend is required for these lines.

5. the manuscript is a little tedious and seems missing the focus. The authors can do more analysis and discuss more on the terpenoids as mentioned in the introduction.

6. PLOS authors have the option to publish the peer review history of their article (what does this mean?). If published, this will include your full peer review and any attached files.

Reviewer #1: No

Reviewer #2: **Yes: **Mohsen Niazian

Reviewer #3: No

---

## [Author Response · Author response to Decision Letter 0]

29 Nov 2020

Response to Reviewers

Journal Requirements:

2.Thank you for stating the following in the Financial Disclosure section:

[Our gratitude goes to the Department of Science and Technology, South Africa, for the Biocatalysis Initiative. MST received a PhD bursary from the National Research Foundation, South Africa, grant number ]. 

We note that one or more of the authors are employed by a commercial company: Omics Center Graz, BioTechMed and ACIB GmbH

 Sections for Financial Disclosure, Funding Statement and Competing Interests Statement have been included in the cover letter.

The “data not shown” statement has been replaced with “supplementary heatmap data and annotation table” since the data is provided in the respective supplementary material.

Reviewers' comments:

Reviewer's Responses to Questions

Comments to the Author

1. Is the manuscript technically sound, and do the data support the conclusions?

Reviewer #1: Yes

Reviewer #2: Yes

Reviewer #3: Yes

2. Has the statistical analysis been performed appropriately and rigorously?

Reviewer #1: No

Reviewer #2: Yes

Reviewer #3: Yes

3. Have the authors made all data underlying the findings in their manuscript fully available?

Reviewer #1: Yes

Reviewer #2: Yes

Reviewer #3: Yes

4. Is the manuscript presented in an intelligible fashion and written in standard English?

Reviewer #1: Yes

Reviewer #2: Yes

Reviewer #3: Yes

5. Review Comments to the Author

Reviewer #1: The research question is well defined, and I believe the experiment was well designed and executed properly. However, there are a few issues, as highlighted in the basic reporting section, where some information should be clarified, or extra information should be included. In particular, please be more transparent and replicates, comparisons, and use of the same/different tissue for the qPCR.

Validity of the findings

I believe this experiment to have been executed rigorously, but the manuscript needs more detail to ensure transparency about the number of plants/replicates/trials done.

1 The differential expressed genes must be validated using qRT-PCR, please added this part correctly. Because I noticed that the author only used two replicates for transcription sequencing.

RNA-seq has become an extremely efficient technology for quantifying relative transcript abundance and studies have shown that its accuracy rivals that of other methods such as microarrays and qRT-PCR, thus precluding the need for further validation (please see Su et al, 2014 below). 

Nevertheless, in this study, validation was carried out at a proteomic level. We identified 328 proteins out of the 956 upregulated transcripts. This represents validation at a much larger scale than qRT-PCR can be performed at. Unfortunately, we are unable to perform additional qRT-PCR experiments at this point due to limited resources. However, we believe that these additional experiments will not add any value to this study given that proteomic validation was already performed. 

Su, Z., Łabaj, P. P., Li, S., Thierry-Mieg, J., Thierry-Mieg, D., Shi, W., ... & Jones, W. D. (2014). A comprehensive assessment of RNA-seq accuracy, reproducibility and information content by the Sequencing Quality Control Consortium. Nature Biotechnology, 32(9), 903-914

2 Association analyzed must be conducted between the differential expressed genes (DEGs) and the differential expressed proteins (DEPs) identified in your study. Positive? Negative or neither? Please added them and make it clearly.

The differentially expressed transcripts were validated using proteomic analysis. However, the proteomic analysis study was not carried out quantitatively to compare differential protein expression with differential transcript expression. Rather, it was performed to confirm that a large portion of the upregulated transcripts were also found within the proteome. 

This information was discussed in lines 708 to 716 in the Revised manuscript with Track Changes: 

“Having identified 3,927 unique proteins and 143,549 transcripts across the three tissues, this producing a proportion of one protein to every ~36.6 transcripts (or 2.7%) at a global level. However, this ratio is much higher when comparing the number of proteins identified within the upregulated transcripts with the number of upregulated transcripts. That is, 328 protein matches out of 956 upregulated transcripts resulting in one protein for every ~2.9 upregulated transcripts (or 34.7%) (Figs 3 and 4). This indicates that the proteomic profiling is reflective of the upregulated transcripts identified within the 3 tissues and serves to validate to a large extent the differentially expressed transcripts identified in Fig 4.”

Basic Reporting

Abstract and Introduction

Overall the information presented in the abstract and introduction is relevant and interesting and does a good job at describing the background and the biological question. However, the English is not very clear in places and therefore the manuscript would benefit from proofreading and correcting in places.

For example:

1 • Line 5-7: “The metabolites of H. hemerocallidea have been identified in several studies. More recently, the terpenoids of the plant have been identified .However, the biochemical pathways and the enzymes involved in the production of metabolites have not been characterised..”

This sentence is incompletely and needed to make clearly.

Line 5-7 has been rephrased to make the meaning clear. 

The sentences now read as:

‘The metabolites contributing to the medicinal properties of H. hemerocallidea have been identified in several studies and, more recently, the active terpenoids of the plant were profiled. However, the biosynthetic pathways and the enzymes involved in the production of the terpene metabolites in H. hemerocallidea have not been characterised at a transcriptomic or proteomic level.’

Materials and methods

The level of detail is sufficient but the English needs revision.

1 •Line 84-85 “Flower and leaf material were immediately frozen in liquid nitrogen upon collection and stored at -80 ºC until use”.

Please add the information how to collected flower and leaf material? How many plants? How many days after planting and How to keep the plant? as well as how many replicates? Make all of them carefully and clearly.

Information on how to collect the flower and leaf material, collection time from planting (not in days as that information is not available), natural growing conditions, and whether specimens were collected as biological replicates or technical replicates was added.

The paragraph now reads as (lines 91 to 100 in Revised Manuscript with Track Changes): 

“An approximately two-year-old Hypoxis hemerocallidea (African potato) plant, grown under natural conditions, was identified and collected at the Pretoria National Botanical Gardens (South Africa) under the expertise of Dr. Robert Archer. Biological replicates of the flower and leaf material were cut and stored in 50 ml centrifuge tubes. The flower material included the sepal and receptacle but not the stamen. The samples were immediately frozen in liquid nitrogen and stored at -80 ºC until use. The corm was washed with distilled water, cut into cubes of approximately 2 cm3, frozen in liquid nitrogen and stored at -80 ºC until use. The lack of H. hemerocallidea specimens, restricted the experimental setup to technical replicates for the corm tissue. Unless otherwise stated, plant material was routinely crushed into a fine powder in liquid nitrogen using a sterile mortar and pestle.”

2 •Line 118-120 “The high-quality paired-end reads of the flower, leaf and corm tissue of H. hemerocallidea were concatenated and assembled de novo (in the absence of a reference genome) into a single RNA-seq dataset using Trinity version 2.6.6 under default settings”

How to eliminate for mutli-duplication’s reads, how to obtain unigene, how to assemble the contigs into one transcript, you must be make it clearly.

The methodology for de novo assembly was carried out under default settings using the Trinity software. As such, it is beyond the scope of this study to explain the presence and elimination of multi-duplication reads or the production of unigenes by the Trinity software from the multiple contigs assembled by the software. These technical aspects of the Trinity software are presented in reference 24 and cannot be reviewed here because it falls outside the scope of the study.

3 In section 153 Differential transcript expression, it is not clear what comparisons have been made. Is it any comparisons were made? Also please describe the methodology in more details, particularly with respect to replicates.

please state how you did the clustering A lot more detail is needed for this part of the analysis.

A comparison between the corm, leaf and flower tissues of H. hemerocallidea was made as outlined in the Differential expression analysis section: “ Differential transcript expression analysis between the corm, leaf and flower tissues of H. hemerocallidea was performed using edgeR [40] to identify transcripts expressed in significantly elevated levels which could possibly confer some of the phytomedicinal associated with the corm and leaf tissues.”

Hierarchical clustering is a standard procedure within the Trinity workflow implemented to depict and finalise the differential expression analysis. To elaborate on that, additions were made to the methodology in the manuscript; lines 175 to 187 in the Revised Manuscript with Track Changes:

“Alignment-based isoform abundance estimation was performed using RNA-Seq by Expectation Maximization (RSEM) version 1.3.1 [42]. That is, the raw reads of each tissue were aligned with Bowtie to the assembled transcriptome. The relative abundance of transcripts in each tissue was estimated with RSEM. Differential transcript expression analysis between the corm, leaf and flower tissues of H. hemerocallidea was performed using edgeR [43] with log2(fpkm+1) normalisation. Differential expression analysis was performed to identify transcripts expressed in significantly elevated levels which could possibly confer some of the phytomedicinal properties associated with the corm and leaf tissues. The workflow was implemented using the Trinity pipeline [27] with a cut-off p-value set to 0.05. Clustering of expressed transcripts between the sample replicates from the leaf and flower and technical replicates from the corm tissue was performed hierarchically using normalised transcript expression obtained using edgeR. Transcripts are clustered together by similar expression levels.”

Results

1. 452 Differential transcript expression

The comparison of the reads to other species is not clear – it is missing from the materials and methods section. Please clarify the reason for these species being used, how they were compared.

The comment made by the reviewer is difficult to place in context because there was no mention made of different species being compared at a differential transcript expression level within the manuscript submitted for revision. 

To attempt to answer the reviewer’s question, an assumption is made that the reviewer is referring to the taxonomic distribution of the annotated transcripts. The following text was added to clarify the reason for using certain species to compare the sequence similarity between species. Lines 165 to 172 in the Revised Manuscript with Track Changes:

“Hypoxis hemerocallidea is classified under the Asparagales order. However, there were no species from the Asparagales order present within the top 10 similar species. To get an insight into the evolutionary relatedness of H. hemerocallidea to species from the Asparagales order a blastn search was performed against the transcriptomes of Asparagus officinalis (garden asparagus) (RefSeq: GCF_001876935.1), Dendrobium catenatum (RefSeq: GCA_001605985.2 and Phalaenopsis equestris (RefSeq: GCF_001263595.1) (from the Asparagales order). The top 6 similar species were included in the analyses as well.” 

2.Please add the part of “Verification of the gene expression profiles of candidate DEGs by qRT-PCR” and “Statistical analysis” In methods and results part. Please denote the association analyzed between DEGs and DEPs identified in this research.

A detailed explanation of why qRT-PCR was not performed is provided in a previous response to this reviewer (please see comment 1). 

To address the second part of the reviewer’s request: “Please denote the association analyzed between DEGs and DEPs identified in this research.”: 

To numerous extents, associations have been made (“denoted”) in the manuscript between differentially expressed transcripts and proteins detected within the tissues. For example, Figures 5 and 6 are clearly relating differentially expressed transcripts and expressed proteins. 

Discussion

Content in the discussion is good and is a relevant discussion of the results. However, the English is poor and therefore would benefit from proofreading.

The manuscript has been proofread and edited appropriately. 

Reviewer #2: Manuscript present transcriptome and the proteome sequencing of medicinal African potato (Hypoxis hemerocallidea). Numerous terpene synthases were identified through functional annotation. Differential expression analysis showed that which tissue upregulateslinalool synthase.

It is valuable, because combined transcriptome and proteome analyses gives a complete insight of genetics and biochemistry of this valuable medicinal plant.

There are some comments that must be addressed by authors before possible acceptance of the manuscript:

Abstract:

- Lines 2-7: This paragraph is belongs to Introduction. Just one line is abstract is enough to shows the importance of plant of interest.

Reviewer 1 recommended the expansion of lines 5-7 for clarity purposes. We chose to adhere to the guidelines suggested by Reviewer 1 because this also provides additional background to why this study was carried out which may be useful to some readers. Moreover, the abstract adheres to the word limit of the journal therefore removal of sentences is not necessary. 

- Line 14: … such as nerolidol synthase, germacrene D synthase, ….

A comma has been added after germacrene D synthase. 

Introduction:

- Line 26: Hypoxidaceae must be written in Italic.

Hypoxidaceae is a plant family name which is not italicised. (See reference below)

(For reference please see: Nelson, L. S., & Balick, M. J. (2020). Section 1. Botanical Nomenclature and Glossary of Botanical Terms. In Handbook of Poisonous and Injurious Plants (pp. 1-18). Springer, New York, NY.)

- Line 31: Please change “medical conditions” with an appropriate phrase.

“medical conditions” changed to “illnesses”

- Line 32: … some cancers, and HIV-AIDS.

A comma has been added after ‘some cancers’. 

- Line 35: Reference “(Liebenberg, 1969)” is out of the instruction of journal. Please keep the same format for references.

Citation was corrected to [3] instead of “(Liebenberg, 1969)”.

- Line 36: Latter research has corroborated some of the medicinal properties (Reference??).

References 4,5 and 6 are in text following the specific examples of corroborated medicinal properties. (Lines 39 to 43 in Revised Manuscript with Track Changes).

- Line 47: please rewrite “and so have their use in the treatment of inflammation”.

The phrase “and so have their use in the treatment of inflammation” has be re-written as “alongside their use in the treatment of inflammation”. (Line 52 of Revised Manuscript with Track Changes).

- Line 51-52: please mention the reasons why “the cultivation of H. hemerocallidea is known to be problematic”??

The sentence was extended as follows to provide a reason for difficulty in cultivation: “the cultivation of H. hemerocallidea is known to be problematic due to lengthy seed dormancy”. (Line 56 of Revised Manuscript with Track Changes)

- Line 49-52: Author mentioned that (i) synthetic production of hypoxoside is difficult, (ii) tissue culture of H. hemerocallidea produces low yields of hypoxoside rendering this method impractical, and (iii) the cultivation of H. hemerocallidea is known to be problematic. With this testimonial, what solution is provided by the authors?

The sentence: “This leaves an alternative, important and yet unexplored gap in the biocatalytic production of hypoxoside, rooperol and other important metabolites which can circumvent the need for cultivation or tissue culturing of Hypoxis hemerocallidea” was placed in text in sequence after the testimonial indicated by the reviewer. The text suggests a solution that biocatalytic alternatives could be adapted to produce the metabolites of interest. (Lines 62 and 63 in the Revised Manuscript with Track Changes).

- Line 60: Please give some examples of “ other plant sources”.

The sentence was rephrased as follows: “However, this compound can be produced on an industrial scale from Chlorophytum borivilianum” to provide a specific example. (Line 65 in the Revised Manuscript with Track Changes).

- Line 60: highly unlikely??!

Rephrased to “unlikely” (Line 66 in the Revised Manuscript with Track Changes).

- Line 72: …. H. hemerocallidea was …

Rephrased to: “…H. hemerocallidea was…” (Line 77 in the Revised Manuscript with Track Changes).

- Introduction section needs some examples of similar researches in other medicinal plants.

To address the request of the reviewer, the following text was added in text with some editing to text already present:

Lines 81 to 86 of Revised Manuscript with Track Changes:

“Cross-tissue transcriptomic analyses have been performed in the past to identify candidate genes involved in the biosynthesis of secondary metabolites in medicinal plants like Ferula asafoetida, Dysphania schraderiana and Salvia miltiorrhiza [21–23]. In this study, aside from transcriptomic analyses, proteomic profiling was performed as well on the flower leaf and corm tissues from Hypoxis hemerocallidea.”

Methods and materials

- Line 84: Flower and leaf materials …

In response to comments from Reviewer 1, the sentence was rephrased, and more information was added. For the sentence to remain grammatically correct, ‘material’ will not be changed to ‘materials’. 

The sentence now reads as: 

‘Biological replicates of the flower and leaf material were cut and stored in 50 ml centrifuge tubes.’ (Line 93 and 94 in the Revised Manuscript with Track Changes).

There is no additional comments related to other sections of “Methods and materials” and I appreciate the authors’ efforts for complete writing and explanations of these sections.

Results

- Line 357: ORFs: Please mention the abbreviations in their first use in the text.

The term “Open reading frame” was mentioned in full on line164 of the Revised Manuscript with Track Changes within the methodology section for the first time and it is not needed to re-mention here.

Discussion

- Lines 633-650: Taxonomic distribution of annotated transcripts: there is no discussion for this paragraph and it seems that it is a “Results” section not “Discussion”.

The Taxonomic distribution of annotated transcripts was discussed. Please see lines 667 to 683 in the Revised Manuscript with Track Changes.

- Lines 652-665: Paragraph “Differential expression”. It also similar to “Results” section not “Discussion”.

The Differential expression was discussed. Please see lines 686 to 698 in the Revised Manuscript with Track Changes.

- Lines 685-708; Paragraph “Terpenoid biosynthesis” there is no related references in this section.

There are not many studies to reference terpenoid profile of the African potato to at a specific level. However, references to multi-product terpenoid synthases has been made [55] as well as a reference to the metabolomic study of the African potato performed in [19]. (Line 735 and 740 in the Revised Manuscript with Track Changes).

Reviewer #3: The authors sequenced the transcriptome, proteome of three tissues of Hypoxis hemerocallidea (African potato), and did the assembly and functional annotation of the transcriptome, different expression analysis of the transcripts, cross-analysis of transcriptome and proteome, and analyzed the secondary metabolism.

I have the following major concerns:

1. The authors did decontamination on the contigs after the assembly. I suggest running decontamination on the fastq files before the assembly. the reads from the other species can cause assembly errors, and the contigs can contain a mix of the reads from different species. Maybe that is why the assembly is too fragmental.

The tool used in decontaminating the transcriptome assembled here was DeconSeq. It is a post-assembly decontamination tool published in a peer-reviewed journal (For reference see: Schmieder, R., & Edwards, R. (2011). Fast identification and removal of sequence contamination from genomic and metagenomic datasets. PloS one, 6(3), e17288.). The implementation of the tool was carried out as recommended. A pre-assembly decontamination procedure was not feasible due to the unavailability of the raw data of the contaminating species.

2. I don't recommend the authors run assembly by default parameters on Trinity and then stopped to refine the assembly results. The assembly results are the cornerstone of the whole project, better assembly results can avoid analysis errors. 

The major component of data refinement was decontamination which could not be avoided or performed pre-assembly to justify optimisation of the assembly process. Furthermore, the default parameters are also suggested by the creators of the Trinity assembler.

Some minor comments:

1. the last row of table 1 is not well described, especially the last item (n: 3278)

The “n:” refers to the number of BUSCO groups searched. The phrase “…after searching 3,278 BUSCO groups…” was added to line 374 in the Revised Manuscript with Track Changes for clarification. 

2. for fig 3, set a title to the two plots. the plot on the right is not suitable for the line plot. because the data are independent of each other.

Figure 3 has a legend; a title will only add redundancy to the needed figure legend. Therefore, we opt for a diagram without a title. The second part of the reviewer’s comment is unclear. The expression level is dependent on the tissue type and the lines depict the expression level of all samples and tissues in for each cluster. The data is dependent. For clarity, the following sentence was added to the Fig 3 legend: “Line plots depict the expression levels of transcripts for each of the three clusters.”

3. at line 324-325, the authors say "A total of 35,087,914 fragments (S2 Fig) were then assembled de novo with Trinity.", but the Fig S2 title says "number of trimmed fragments", it is a little confusing, I suggest changing it as "number fragments after trimmed".

The phrase “…A total of 35,087,914 fragments (S2 Fig)…” was changed to “…A total of 35,087,914 trimmed fragments (S2 Fig)…”. (Lines 354 and 355 in the Revised Manuscript with Track Changes).

4. There is a yellow line in the raw reads plot in Fig S1, while the others are red. A legend is required for these lines.

The following sentence was added to the legend of S1 Fig: “The colour of the lines depicts the quality of the reads. Green indicates good quality; orange indicates medium quality and red indicates low quality.”

5. the manuscript is a little tedious and seems missing the focus. The authors can do more analysis and discuss more on the terpenoids as mentioned in the introduction.

We believe we have derived valuable information from the analyses as it stands and we have depicted these data in an appropriate way in a single figure (Fig 6) which makes it easy for researchers to understand the findings of this study.

6. PLOS authors have the option to publish the peer review history of their article (what does this mean?). If published, this will include your full peer review and any attached files.

Do you want your identity to be public for this peer review? For information about this choice, including consent withdrawal, please see our Privacy Policy.

Reviewer #1: No

Reviewer #2: Yes: Mohsen Niazian

Reviewer #3: No

---

## [Decision Letter · Decision Letter 1]

18 Dec 2020

PONE-D-20-28875R1

Transcriptome and proteome of the corm, leaf and flower of Hypoxis hemerocallidea (African potato)

PLOS ONE

Dear Dr. Rumbold,

Thank you for submitting your manuscript to PLOS ONE. After careful consideration, we feel that it has merit but does not fully meet PLOS ONE’s publication criteria as it currently stands. Therefore, we invite you to submit a revised version of the manuscript that addresses the points raised during the review process.

We look forward to receiving your revised manuscript.

Kind regards,

Xiang Jia Min, Ph. D.

Academic Editor

PLOS ONE

Journal Requirements:

Additional Editor Comments (if provided):

Reviewers' comments:

Reviewer's Responses to Questions

**Comments to the Author**

1. If the authors have adequately addressed your comments raised in a previous round of review and you feel that this manuscript is now acceptable for publication, you may indicate that here to bypass the “Comments to the Author” section, enter your conflict of interest statement in the “Confidential to Editor” section, and submit your "Accept" recommendation.

Reviewer #1: All comments have been addressed

Reviewer #2: All comments have been addressed

Reviewer #3: (No Response)

2. Is the manuscript technically sound, and do the data support the conclusions?

Reviewer #1: Yes

Reviewer #2: Yes

Reviewer #3: Yes

3. Has the statistical analysis been performed appropriately and rigorously? 

Reviewer #1: N/A

Reviewer #2: Yes

Reviewer #3: Yes

4. Have the authors made all data underlying the findings in their manuscript fully available?

Reviewer #1: Yes

Reviewer #2: Yes

Reviewer #3: Yes

5. Is the manuscript presented in an intelligible fashion and written in standard English?

Reviewer #1: Yes

Reviewer #2: Yes

Reviewer #3: Yes

6. Review Comments to the Author

Reviewer #1: All comment has been addressed. I think it would be suitable for publish after carefully checked.

I verify that all required questions have been answered and that all responses meet formatting specifications.

Reviewer #2: (No Response)

Reviewer #3: My two major concerns are not addressed.

1. I recommend decontamination before assembly because the contamination will mess up your assembly. Since the authors used DeconSeq, it can be used on fastq files before assembly. Furthermore, in the DeconSeq paper, the authors talking about it in the first paragraph,  "Those sequence contaminations are a serious concern to the quality of the data used for downstream analysis, causing misassembly of sequence contigs and erroneous conclusions. Therefore, the removal of sequence contaminants is a necessary and required step for all sequencing projects." I think you misused this software.

2. The Trinity paper provides an extra operation box (box 4) to describe the parameters for you to tune your parameters, and the authors never said the default parameters are suggested, the authors said " Users can include additional parameter settings (see below) to tune any of the three assembly steps according to the characteristics of the dataset, but Trinity usually performs well with the default parameters." Your data contains contamination and you should be more cautious on the parameters.

7. PLOS authors have the option to publish the peer review history of their article (what does this mean?). If published, this will include your full peer review and any attached files.

Reviewer #1: No

Reviewer #2: **Yes: **Mohsen Niazian

Reviewer #3: No

---

## [Author Response · Author response to Decision Letter 1]

29 Jan 2021

Comments to the Author

1. If the authors have adequately addressed your comments raised in a previous round of review and you feel that this manuscript is now acceptable for publication, you may indicate that here to bypass the “Comments to the Author” section, enter your conflict of interest statement in the “Confidential to Editor” section, and submit your "Accept" recommendation.

Reviewer #1: All comments have been addressed

Reviewer #2: All comments have been addressed

Reviewer #3: (No Response)

2. Is the manuscript technically sound, and do the data support the conclusions?

Reviewer #1: Yes

Reviewer #2: Yes

Reviewer #3: Yes

3. Has the statistical analysis been performed appropriately and rigorously?

Reviewer #1: N/A

Reviewer #2: Yes

Reviewer #3: Yes

4. Have the authors made all data underlying the findings in their manuscript fully available?

Reviewer #1: Yes

Reviewer #2: Yes

Reviewer #3: Yes

5. Is the manuscript presented in an intelligible fashion and written in standard English?

Reviewer #1: Yes

Reviewer #2: Yes

Reviewer #3: Yes

6. Review Comments to the Author

Reviewer #1: All comment has been addressed. I think it would be suitable for publish after carefully checked.

I verify that all required questions have been answered and that all responses meet formatting specifications.

Reviewer #2: (No Response)

Reviewer #3: My two major concerns are not addressed.

1. I recommend decontamination before assembly because the contamination will mess up your assembly. Since the authors used DeconSeq, it can be used on fastq files before assembly. Furthermore, in the DeconSeq paper, the authors talking about it in the first paragraph, "Those sequence contaminations are a serious concern to the quality of the data used for downstream analysis, causing misassembly of sequence contigs and erroneous conclusions. Therefore, the removal of sequence contaminants is a necessary and required step for all sequencing projects." I think you misused this software.

We disagree with the reviewer. A more careful scrutiny of the paper is required in order to understand why the tool was used post assembly.

DeconSeq is a tool that removes contaminating sequences based on sequence similarity. The user’s dataset is aligned to a ‘remove’ dataset which contains genomic or transcriptomic sequences of the contaminating species. Sequences are then classified as contamination and removed if they match any sequence on the ‘remove’ database above a set threshold value. 

However, the authors clearly state that the tool has been designed for longer-read datasets. And since removal is based on sequence similarity, the tool is unable to accurately identify contamination if sequences are anything less than 150bp. Please see what the authors state below: 

“Here, we selected longer-read alignment programs that are actively maintained and widely used and evaluated them on BLAST+ simulated datasets. These programs include BLAST, BLAST+ [37], Mosaik, NUCmer (from MUMmer package), and BWA-SW. Based on the evaluation results, we adopted BWA-SW for the removal of human sequence contamination from metagenomes and developed DeconSeq, a robust framework for the rapid, automated identification and removal of sequence contamination from longer-read datasets.”

“The simulated data contained 200 bp, 500 bp, or 1,000 bp long sequences. Errors were introduced at rates of 2% and 5%. The typical error rate for real data is approximately 0.5%, therefore this analysis provides a worst-case scenario [38].”

“Using the default settings, longer sequences could be aligned to the correct region more often than shorter sequences independent of the error rates introduced.”

“Variation in read length did affect the accuracy of DeconSeq in identifying contaminating sequences, as mainly short sequences were misclassified.”

Sequencing on the Illumina platform produces reads of 125bp and adaptor sequences have to then be trimmed, leaving the raw reads with a length of around 50bp. Other sequencing technologies such as 454 sequencing produces longer raw reads (300-500bp) and therefore DeconSeq may be used on the raw reads. But given the sequence length requirements for DeconSeq, this is not possible for Illumina reads and therefore assembly is necessary prior to decontamination. The authors do not infer that the tool can only be used on raw reads prior to assembly. They state that the input is a FASTA file. Hence, conditions for a particular dataset must be decided based on application of the data and sequencing technology used. Please see what the authors state below: 

“The ability to map sequence reads uniquely to the correct location is dependent on a number of factors such as the complexity of the reference data (highly polymorphic or repetitive regions), length of the sequence reads, error rates of the reads, and the diversity of the individual organism compared to the reference[23,48].”

“It is more likely to find sufficient seeds from which to extend the alignment for longer reads. As read length increases, the mapping in repetitive regions will improve. We showed that the BWA-SW program used by DeconSeq has a high sensitivity (including repetitive regions) for sequences with low error rates or longer reads when aligning human DNA to the reference genome.”

“The choice of the alignment program depends on the biological application and on the type of sequencing technology used to generate the data.”

“The metagenomes used in this study were pre-processed prior to any processing with DeconSeq. UniVec build 5.2 (http://www.ncbi.nlm.nih.gov/VecScreen/UniVec.html) and cross_match (http://www.phrap.org/) were used to screen for vector contamination in the metagenomes. TagCleaner [54] was used to trim adapter and tag sequences. PRINSEQ [52] was then used to filter exact sequence duplicates, sequences shorter than 50 bp or longer than 10,000 bp, sequences containing more than 5% of ambiguous base N after trimming Ns from the sequence ends, and sequences containing non IUPAC conform characters for DNA sequences. The resulting datasets were excluded from the study if the mean sequence length was below 150 bp or the dataset contained less than 1,000 metagenomic sequences. Metagenomes targeted to single loci such as 16S rRNA studies were excluded as well.”

In this manuscript, the Illumina HiSeq2500 platform was used. In this case, if the raw reads were used for decontamination, DeconSeq would most certainly misclassify sequences due to inaccurate alignments. Following assembly with Trinity, transcripts in this study ranged from 201bp to 5 874bp which then allowed for sequences to be compatible with the requirements for DeconSeq. A total of 74 652 Helianthus annuus sequences were removed from the dataset. We even went on further to validate the removal of Helianthus annuus sequences by producing a similarity profile with the contaminating species before and after decontamination (S3 Fig). This undoubtedly confirmed the removal of Helianthus annuus sequences. 

Post-assembly decontamination of datasets is a highly accepted method for decontamination and numerous post-assembly decontamination tools currently exist. In a recent study published in Bioinformatics, various post-assembly decontamination tools were compared and DeconSeq was one of the post-assembly decontamination tools assessed. DeconSeq can be used post-assembly since removal of contamination is based on sequence similarity. The study first assembled numerous transcriptomes with Trinity (default parameters) and then went on to test decontamination using various tools - including DeconSeq [1].

In fact, we performed a literature search to confirm that the tool was not misused. We found hundreds of papers that utilize DeconSeq after assembly. For the purpose of this rebuttal, we chose to only focus on similar studies published in PLOS One. Of the 27 papers published in PLOS One that used DeconSeq, 12 papers utilised the tool post-assembly like we did [2–13]. If one reads these articles, a clear trend emerges. Papers that used sequencing technologies which produce longer raw reads, used the tool before assembly. While papers that used the Illumina sequencing technology, assembled first and then used the tool post-assembly. We therefore disagree that DeconSeq must be used before the assembly and no changes were made to the manuscript. 

2. The Trinity paper provides an extra operation box (box 4) to describe the parameters for you to tune your parameters, and the authors never said the default parameters are suggested, the authors said " Users can include additional parameter settings (see below) to tune any of the three assembly steps according to the characteristics of the dataset, but Trinity usually performs well with the default parameters." Your data contains contamination and you should be more cautious on the parameters.

The default parameters in Trinity have been optimised to give the best results with most datasets. And this is why Trinity is routinely used with default parameters. This question has been raised before to the developers of Trinity on GitHub, where the source code for Trinity is stored and frequently updated (https://github.com/trinityrnaseq/trinityrnaseq/issues/342). The developer of Trinity clearly suggests that the default Trinity parameters are usually fine for most purposes. Please see what the author states below: 

“Using the default Trinity parameters for doing an assembly should be fine for most purposes. I would suggest reassembling only if you have additional rna-seq data to add to your current data set. I'd suggest exploring other alternative assemblers as well in case there's another one that performs better on your data for whatever reason.”

However, exploring different parameters or even assemblers falls beyond the scope of this study. Reassembly and re-annotation represents a large portion of work, and redoing the analysis may not necessarily guarantee better results. Furthermore, and most importantly, we are unable to repeat the assembly and annotation at this point in time with the resources that we currently have.

It was important for both the contaminants and the African potato reads to assemble correctly for successful decontamination. We believe that the default Trinity parameters served this objective well and that we have derived valuable information from the analysis as it stands. This is evidenced by the fact that we managed to remove 74 652 contaminating Helianthus annuus sequences, with similarity plots confirming successful removal. In addition, we achieved a 47.5% annotation rate for the African potato, falling well within the expected annotation rates for plants without reference genomes suggesting successful assembly. Moreover, the proteomic analysis of the African potato tissues mapped to the transcriptome with significant success. Thus, we concluded that we have proceeded correctly to decontaminate and assemble the transcriptome. 

In this instance, a simple literature search resulted in thousands of papers published in various journals that used Trinity with default settings. However, for the purpose of this rebuttal, we chose to focus on papers that utilised both Trinity and DeconSeq and that were published in PLOS one. From the 12 articles that used DeconSeq post-assembly, 8 of these articles also used Trinity with default parameters [2–7,11,12]. This increasingly suggests that our analysis is technically sound. 

References

1. Lafond-Lapalme J, Duceppe M-O, Wang S, Moffett P, Mimee B. A new method for decontamination of de novo transcriptomes using a hierarchical clustering algorithm. Bioinformatics. 2017;33: 1293–1300. doi:10.1093/bioinformatics/btw793

2. Carlson DE, Hedin M. Comparative transcriptomics of Entelegyne spiders (Araneae, Entelegynae), with emphasis on molecular evolution of orphan genes. PLOS ONE. 2017;12: e0174102. doi:10.1371/journal.pone.0174102

3. Kwon CW, Park K-M, Kang B-C, Kweon D-H, Kim M-D, Shin SW, et al. Cysteine Protease Profiles of the Medicinal Plant Calotropis procera R. Br. Revealed by De Novo Transcriptome Analysis. PLOS ONE. 2015;10: e0119328. doi:10.1371/journal.pone.0119328

4. Salvemini M, Arunkumar KP, Nagaraju J, Sanges R, Petrella V, Tomar A, et al. De Novo Assembly and Transcriptome Analysis of the Mediterranean Fruit Fly Ceratitis capitata Early Embryos. PLOS ONE. 2014;9: e114191. doi:10.1371/journal.pone.0114191

5. Belleghem SMV, Roelofs D, Houdt JV, Hendrickx F. De novo Transcriptome Assembly and SNP Discovery in the Wing Polymorphic Salt Marsh Beetle Pogonus chalceus (Coleoptera, Carabidae). PLOS ONE. 2012;7: e42605. doi:10.1371/journal.pone.0042605

6. Hyun TK, Lee S, Rim Y, Kumar R, Han X, Lee SY, et al. De-novo RNA Sequencing and Metabolite Profiling to Identify Genes Involved in Anthocyanin Biosynthesis in Korean Black Raspberry (Rubus coreanus Miquel). PLOS ONE. 2014;9: e88292. doi:10.1371/journal.pone.0088292

7. Benton MA, Kenny NJ, Conrads KH, Roth S, Lynch JA. Deep, Staged Transcriptomic Resources for the Novel Coleopteran Models Atrachya menetriesi and Callosobruchus maculatus. PLOS ONE. 2016;11: e0167431. doi:10.1371/journal.pone.0167431

8. Agunbiade TA, Sun W, Coates BS, Djouaka R, Tamò M, Ba MN, et al. Development of Reference Transcriptomes for the Major Field Insect Pests of Cowpea: A Toolbox for Insect Pest Management Approaches in West Africa. PLOS ONE. 2013;8: e79929. doi:10.1371/journal.pone.0079929

9. Erlandson MA, Mori BA, Coutu C, Holowachuk J, Olfert OO, Gariepy TD, et al. Examining population structure of a bertha armyworm, Mamestra configurata (Lepidoptera: Noctuidae), outbreak in western North America: Implications for gene flow and dispersal. PLOS ONE. 2019;14: e0218993. doi:10.1371/journal.pone.0218993

10. Leese F, Brand P, Rozenberg A, Mayer C, Agrawal S, Dambach J, et al. Exploring Pandora’s Box: Potential and Pitfalls of Low Coverage Genome Surveys for Evolutionary Biology. PLOS ONE. 2012;7: e49202. doi:10.1371/journal.pone.0049202

11. Sayadi A, Immonen E, Bayram H, Arnqvist G. The De Novo Transcriptome and Its Functional Annotation in the Seed Beetle Callosobruchus maculatus. PLOS ONE. 2016;11: e0158565. doi:10.1371/journal.pone.0158565

12. Tao T, Zhao L, Lv Y, Chen J, Hu Y, Zhang T, et al. Transcriptome Sequencing and Differential Gene Expression Analysis of Delayed Gland Morphogenesis in Gossypium australe during Seed Germination. PLOS ONE. 2013;8: e75323. doi:10.1371/journal.pone.0075323

13. Kuo RC, Zhang H, Zhuang Y, Hannick L, Lin S. Transcriptomic Study Reveals Widespread Spliced Leader Trans-Splicing, Short 5′-UTRs and Potential Complex Carbon Fixation Mechanisms in the Euglenoid Alga Eutreptiella sp. PLOS ONE. 2013;8: e60826. doi:10.1371/journal.pone.0060826

---

## [Decision Letter · Decision Letter 2]

10 Feb 2021

PONE-D-20-28875R2

Transcriptome and proteome of the corm, leaf and flower of Hypoxis hemerocallidea (African potato)

PLOS ONE

Dear Dr. Rumbold,

Thank you for submitting your manuscript to PLOS ONE. After careful consideration, we feel that it has merit but does not fully meet PLOS ONE’s publication criteria as it currently stands. Therefore, we invite you to submit a revised version of the manuscript that addresses the points raised during the review process.

One of the reviewers still has some questions and concerns regarding decontamination of your data. Please address these questions.

We look forward to receiving your revised manuscript.

Kind regards,

Xiang Jia Min, Ph. D.

Academic Editor

PLOS ONE

Reviewers' comments:

Reviewer's Responses to Questions

**Comments to the Author**

1. If the authors have adequately addressed your comments raised in a previous round of review and you feel that this manuscript is now acceptable for publication, you may indicate that here to bypass the “Comments to the Author” section, enter your conflict of interest statement in the “Confidential to Editor” section, and submit your "Accept" recommendation.

Reviewer #1: All comments have been addressed

Reviewer #3: (No Response)

2. Is the manuscript technically sound, and do the data support the conclusions?

Reviewer #1: Yes

Reviewer #3: Yes

3. Has the statistical analysis been performed appropriately and rigorously? 

Reviewer #1: Yes

Reviewer #3: Yes

4. Have the authors made all data underlying the findings in their manuscript fully available?

Reviewer #1: Yes

Reviewer #3: Yes

5. Is the manuscript presented in an intelligible fashion and written in standard English?

Reviewer #1: Yes

Reviewer #3: Yes

6. Review Comments to the Author

Reviewer #1: It appears that the suggested editing has improved the presentation of the work herein. The supplemental data provided appear to complete requests. it appears that the manuscript is in form to be moved forward; please consider the manuscript accepted.

Reviewer #3: 1. If the authors insist doing assembly after the de-contamination, it raises the following questions:

a. is there any chimera contigs in the assembly results?

b. if yes, how many? Are these contigs been removed by the de-contamination step?

c. if the chimera contigs are removed, how much data/information of the object(African potato) is lost?

d. what is the effect of these data/information lose?

2. is there any proof that run trinity in default parameter is acceptable on data with contaminations like yours?

7. PLOS authors have the option to publish the peer review history of their article (what does this mean?). If published, this will include your full peer review and any attached files.

Reviewer #1: **Yes: **Jian Gao

Reviewer #3: No

---

## [Author Response · Author response to Decision Letter 2]

4 Jun 2021

Comments to the Author

1. If the authors have adequately addressed your comments raised in a previous round of review and you feel that this manuscript is now acceptable for publication, you may indicate that here to bypass the “Comments to the Author” section, enter your conflict of interest statement in the “Confidential to Editor” section, and submit your "Accept" recommendation.

Reviewer #1: All comments have been addressed

Reviewer #3: (No Response)

2. Is the manuscript technically sound, and do the data support the conclusions?

Reviewer #1: Yes

Reviewer #3: Yes

3. Has the statistical analysis been performed appropriately and rigorously?

Reviewer #1: Yes

Reviewer #3: Yes

4. Have the authors made all data underlying the findings in their manuscript fully available?

Reviewer #1: Yes

Reviewer #3: Yes

5. Is the manuscript presented in an intelligible fashion and written in standard English?

Reviewer #1: Yes

Reviewer #3: Yes

6. Review Comments to the Author

Reviewer #1: It appears that the suggested editing has improved the presentation of the work herein. The supplemental data provided appear to complete requests. it appears that the manuscript is in form to be moved forward; please consider the manuscript accepted.

Reviewer #3: 1. If the authors insist doing assembly after the de-contamination, it raises the following questions:

We chose a protocol that was peer reviewed and published in PLOS ONE as well as other journals (which we detailed extensively in our previous response). If these protocols are invalid, these articles should be recalled. However it should not become our responsibility to defend peer-reviewed work that we did not author and is currently published. If the reviewer contests DeconSeq use cases, we suggest taking it up with the relevant authors.

a. is there any chimera contigs in the assembly results?

Perhaps. Other publications using Trinity and Deconseq that we have referred to did not report on chimeras. And as mentioned in previous responses, at this point we do not have the resources to address this. Moreover, there is no available genome of the African potato to compare assembled chimeras with actual chimeras. 

If chimeras resemble Helianthuus annuus sequences more than the similar species of the African potato, they would have been removed by DeconSeq; not because they were chimeras but because they matched better to Helianthus annuus.

b. if yes, how many? Are these contigs been removed by the de-contamination step?

NA

c. if the chimera contigs are removed, how much data/information of the object(African potato) is lost?

NA

d. what is the effect of these data/information lose?

NA

2. is there any proof that run trinity in default parameter is acceptable on data with contaminations like yours?

Yes. We have already provided in the previous response an extensive list of cases of Trinity in conjunction DeconSeq. If the reviewer implies that there should be specific parameters for our unique case, then those other published unique cases should have underwent the same scrutiny, because they are all unique in their own right. If they are faulty, we suggest that PLOS ONE recalls those publications because they have provided misinformation to us. 

7. PLOS authors have the option to publish the peer review history of their article (what does this mean?). If published, this will include your full peer review and any attached files.

Do you want your identity to be public for this peer review? For information about this choice, including consent withdrawal, please see our Privacy Policy.

Reviewer #1: Yes: Jian Gao

Reviewer #3: No

---

## [Editor Report · Decision Letter 3]

14 Jun 2021

Transcriptome and proteome of the corm, leaf and flower of Hypoxis hemerocallidea (African potato)

PONE-D-20-28875R3

Dear Dr. Rumbold,

We’re pleased to inform you that your manuscript has been judged scientifically suitable for publication and will be formally accepted for publication once it meets all outstanding technical requirements.

Kind regards,

Xiang Jia Min, Ph. D.

Academic Editor

PLOS ONE
---

## [Editor Report · Acceptance letter]

29 Jun 2021

PONE-D-20-28875R3 

Transcriptome and proteome of the corm, leaf and flower of *Hypoxis hemerocallidea* (African potato) 

Dear Dr. Rumbold:

I'm pleased to inform you that your manuscript has been deemed suitable for publication in PLOS ONE. Congratulations! Your manuscript is now with our production department. 

Kind regards, 

on behalf of

Dr. Xiang Jia Min 

Academic Editor

PLOS ONE